



# Justification for high ascent attainment for balloon radiosonde soundings at GRUAN and other sites

Masatomo Fujiwara[1], Bomin Sun[2,3], Anthony Reale[3], Domenico Cimini[4], Salvatore Larosa[4], Lori Borg[5], Christoph von Rohden[6], Michael Sommer[6], Ruud Dirksen[6], Marion Maturilli[7], Holger Vömel[8], Rigel Kivi[9], Bruce Ingleby[10], Ryan J. Kramer[11], Belay Demoz[12], Fabio Madonna[13,4], Fabien Carminati[14], Owen Lewis[14], Brett Candy[14], Christopher Thomas[14], David Edwards[14], Noersomadi[15], Kensaku Shimizu[16], and Peter Thorne[17]

[1]Faculty of Environmental Earth Science, Hokkaido University, Sapporo, 060-0810, Japan
[2]I.M. Systems Group, Rockville, Maryland 20852, USA
[3]NOAA NESDIS Center for Satellite Applications and Research (STAR), College Park, Maryland, 20740, USA
[4]National Research Council of Italy (CNR), Institute of Methodologies for Environmental Analysis (IMAA), Potenza, 85050, Italy
[5]Cooperative Institute for Meteorological Satellite Studies (CIMSS), University of Wisconsin-Madison, Madison, WI, 53706, USA
[6]GRUAN Lead Centre, Lindenberg Meteorological Observatory, Deutscher Wetterdienst, Am Observatorium 12, 15848, Tauche, Germany
[7]Alfred Wegener Institute Helmholtz Centre for Polar and Marine Research, Telegrafenberg A45, 14473, Potsdam, Germany
[8]Earth Observing Laboratory, National Center for Atmospheric Research, Boulder, CO, USA
[9]Space and Earth Observation Centre, Finnish Meteorological Institute, Sodankylä, 99600, Finland
[10]European Centre for Medium-Range Weather Forecasts, Shinfield Park, Reading, RG2 9AX, UK
[11]NOAA Geophysical Fluid Dynamics Laboratory, Princeton, NJ, USA
[12]University of Maryland Baltimore County, Baltimore, MD, USA
[13]Department of Physics, University of Salerno, Salerno, 84084, Italy
[14]Met Office, Exeter, EX1 3PB, UK
[15]Research Center for Climate and Atmosphere, National Research and Innovation Agency (BRIN), Bandung, Indonesia
[16]Meisei Electric Co., Ltd., 2223 Naganumamachi, Isesaki, Gunma, 372-8585, Japan
[17]ICARUS Climate Research Centre, Maynooth University, Maynooth, Ireland

*Correspondence to*: Masatomo Fujiwara (fuji@ees.hokudai.ac.jp)

**Abstract.** We assess and illustrate the benefits of high-altitude attainment of balloon-borne radiosonde soundings, up to and beyond 10 hPa level compared to e.g. 30 hPa, at operational stations and at sites of the Global Climate Observing System (GCOS) Reference Upper Air Network (GRUAN). We first discuss technical challenges and the possible solutions for balloon soundings at these higher altitudes. Then, we assess the role of high-ascent radiosonde measurements in climate monitoring and various process studies, contributions to satellite calibration and validation, and impacts on numerical weather prediction systems. The analysis herein shows that the extra costs and technical challenges involved in consistent attainment of high



ascents are more than outweighed by the benefits for a broad variety of real time and delayed mode applications. Consistent attainment of high ascents should therefore be pursued across the GRUAN network and the broader observational network.

## 1 Introduction

Balloon-borne radiosonde soundings represent the longest continuous series of upper-air measurements and still to this day constitute one of the main upper air observation methods, alongside satellite and ground-based remote sensing and aircraft measurements (Chen et al., 2021). Modern upper air soundings using rubber balloons and radiosondes began in the 1920s,

while the proposal of such upper air observations was made already in 1896 by the International Meteorological Organization (IMO) which was succeeded by the World Meteorological Organization (WMO) established in 1951 (e.g. Vömel and Fujiwara, 2021; Edwards, 2010). A good number of sounding data to characterize daily global synoptic weather became available from the late 1940s (e.g. Kalnay et al., 1996; Bell et al., 2021; Kosaka et al., 2024), and data rescue activities are currently still ongoing under the Integrated Global Radiosonde Archive (IGRA) project (Durre et al., 2018), Atmospheric Circulation

Reconstructions over the Earth (ACRE; https://www.met-acre.org/, last access: 25 November 2024), and others (including https://www.ncei.noaa.gov/data/ecmwf-global-upper-air-bufr/, last access: 25 November 2024; see Ingleby et al., 2016 and Geller et al., 2021). Ingleby (2022) made a summary of the status of operational radiosonde reports in 2022. These radiosonde data have been used for research and analysis of the atmosphere (e.g. from synoptic weather to climatology), for operational numerical weather forecasting through data assimilation and verification (e.g. Pauley and Ingleby, 2022), and for climate

change studies and global atmospheric reanalyses (e.g. SPARC, 2022), among other applications. Various activities on homogenization, or bias corrections, for radiosonde temperature data have also been conducted for their use in climate change studies and in reanalyses because of the existence of undocumented instrument-related change points in the data time series at virtually all the sounding sites (e.g. Seidel et al., 2009; Haimberger et al., 2012; Gulev et al., 2021 and the references therein; see also SPARC, 2022, Chapter 2, Section 2.4.3.1). This is because the original motivation for operational radiosonde

measurements was in the short-term weather forecasting and aircraft operations, not necessarily in the long-term monitoring of the atmosphere which needs to detect much smaller changes over time. As explained below, over recent decades, the WMO and other bodies have been establishing improved global radiosonde networks to address this issue.

The launch of the satellite microwave and infrared sounders onboard the Television and Infrared Observation Satellite - Next

generation (TIROS-N) of the National Oceanic and Atmospheric Administration (NOAA) in October 1978 marked the beginning of the "satellite era" for atmospheric monitoring and research (e.g. Spencer et al., 1990; Nash and Brownscombe, 1983; see also SPARC, 2022, Chapter 2, Section 2.4.3.2). Since then, satellite data calibration (CAL) and validation (VAL) have also been important tasks for the radiosonde observations. Note that more recently, numerical weather forecast models





are also used for satellite CAL/VAL, i.e. the models are validated with radiosondes and then are compared with satellite data
(e.g. Newman et al., 2020). Since 2001 (and in particular after 2006), the Global Navigation Satellite System (GNSS) radio
occultation (RO) measurements have been providing temperature profile information in the troposphere and stratosphere at
much higher vertical resolutions than satellite radiance measurements, with long-term stability and small uncertainties (see e.g.
Steiner et al., 2020a and the references therein). For some applications, RO measurements are now more important than those
by the radiosonde network for stratospheric temperatures. In 2008, the NOAA Products validation System was deployed which
routinely compiles collocated radiosonde, satellite (including RO) and selected numerical weather prediction (NWP)
atmospheric profiles daily which provides a key component of the satellite product CAL/VAL and cross-comparisons at
NOAA.  For NWP systems, both radiosondes and RO are used in the variational analysis to "anchor" the bias correction, vital
for the assimilation of satellite radiances (e.g. Eyre, 2016).

Technically it is possible to use balloons for radiosoundings up to altitudes of ~40 km (~3 hPa). Any increase in burst heights
can only be achieved at the expense of increasing costs in terms of balloons and filling gas. The cost-benefit ratio therefore
determines the heights actually achieved in practice. There have been several documents regarding the requirements for height
attainment for balloon-borne radiosonde soundings from operational and research needs. Below, we review some of the recent
ones.

Around the early 1990s, there was a perceived threat to long-term continuity of the global radiosonde network as national
meteorological and hydrological services started closing stations as satellite data and their usage became more prevalent. To
preserve a globally representative subset of the network sufficient to characterise global scale climate variability and changes,
the Global Climate Observing System (GCOS) Upper Air Network (GUAN) was defined in the mid-1990s under the WMO
and other bodies (GCOS, 2002, 2010). GUAN includes stations with long-term, high quality radiosonde observations to
establish an upper-air climate monitoring network. By 2014, GUAN had grown to 170 stations worldwide. GCOS (2010)
provided updated observation requirements for GUAN stations (relative to GCOS, 2002). Regarding the height (pressure)
attainment, GCOS (2010) expresses as follows:

· Minimum requirements (MRQs): Temperature up to 30 hPa; humidity up to the tropopause; and wind direction and speed
up to 30 hPa.

· Target requirements (TRQs) (in addition to the MRQs): Temperature and wind up as high as possible.

More recently, WMO launched the Global Basic Observing Network (GBON, https://community.wmo.int/activity-
areas/wigos/gbon, last access: 25 November 2024) toward "a radical overhaul of the international exchange of observational
data, which underpin all weather, climate and water services and products." Regarding the requirements for upper-air GBON
stations, WMO (2021a) states that "Members shall maintain the continuous operation of a set of upper-air stations/platforms



over land that observe, at a minimum, temperature, humidity and horizontal wind, with a vertical resolution of 100 m or higher, twice a day or better, up to a level of 30 hPa or higher, located such that GBON has a horizontal resolution of 500 kilometres or higher for these observations." and that "Members should operate a subset of the selected GBON upper-air observing

stations/platforms that observe temperature, humidity and horizontal wind up to 10 hPa or higher, at least once per day, located such that, where geographical constraints allow, GBON has a horizontal resolution of 1000 kilometres or higher, for these observations." (Note that 'shall' and 'should' here, as in other WMO regulatory materials, have specific meanings: 'shall' means members must achieve the requirement; whereas 'should' means members are strongly encouraged to achieve the requirement.)


Independent of GUAN and GBON, but with many stations overlapping with these networks, GCOS has been operating the GCOS Reference Upper Air Network (GRUAN) since 2008 (Seidel et al., 2009; Bodeker et al., 2016; https://www.gruan.org/, last access: 25 November 2024). As of November 2024, there are 14 certified GRUAN stations worldwide. GRUAN differs from GUAN and GBON in the following aspects: GRUAN performs reference observations and develops GRUAN data

products (GDPs) based on calibrated raw data. Criteria for the reference quality of the observations are metrological traceability, correction of all known errors and biases, estimates of measurement uncertainties for each data point, and the full transparency for all data processing steps in the documentation. Furthermore, it has a dedicated working group and lead centre; it has a clearly defined certification procedure for the member stations; it holds the implementation and coordination meeting every year or two; and it conducts research through various teams. As of November 2024, there are four radiosonde GDPs

(https://www.gruan.org/data/data-products/gdp, last access: 25 November 2024), namely, RS92-GDP.2 (Dirksen et al., 2014), RS-11G-GDP.1 (Kobayashi et al., 2019; Kizu et al., 2018), RS41-GDP.1 (von Rohden et al., 2022; Sommer et al., 2023), and IMS-100-GDP.2 (Hoshino et al., 2022). Regarding the height attainment requirements for GRUAN sites, GCOS (2007), in its Appendix 1, provides a requirements table, where, for example, the vertical range is specified as "0–50 km" for temperature and pressure (though 50 km is currently unrealistic for in-situ measurements using rubber balloons; however, GRUAN also

considers ground-based remote sensing instruments).

The above requirements for height attainment for balloon-borne radiosonde soundings were provided without explicit reference to a robust scientific rationale, although these requirements had been provided by experts from various atmospheric science fields. Without a robust scientific justification there is a risk that the importance of meeting such targets will be poorly

recognised and thus little effort made to consistently attain such altitudes. The level of 10 hPa (~32 km) has been defined as a target because this marks the lower limit of the maximum altitude range of 10 hPa to 5 hPa (~32 km to 37 km) that can be achieved routinely with rubber balloons of widespread available types and sizes. On the other hand, comprehensive in-situ data from the whole stratosphere and higher layers are highly desired because of their increasing scientific importance. The level of 50 km (~1 hPa) represents the location of the stratopause, and the monitoring of the mesosphere (50 km to 80 km, 1

hPa to 0.01 hPa) may also be very important for climate change studies (e.g. Baldwin et al., 2019). Note also that the model





top of recent global NWP models and global reanalysis systems has been extended to 0.01 hPa (e.g. SPARC 2022, Chapter 2) to fully utilize various satellite data and to appropriately represent the processes in the stratosphere and mesosphere that affect tropospheric weather (e.g. Baldwin et al., 2019). Therefore, in this paper, we aim to provide scientific justifications for radiosonde measurements covering also the 30 hPa to 5 hPa region regularly, by both reviewing relevant publications and

making some new studies. Naturally, this task should be done in comparison with other observing system components including satellite observations. We believe that this paper will also be a useful summary of the state of our atmospheric observing systems in the early 2020s.

In the following, we first discuss technical issues for balloon sounding, i.e. both balloons and radiosonde sensors, and how to

solve them (Section 2). Then, we summarize scientific justifications to attain 10 hPa to 5 hPa pressure altitudes regularly for balloon observations, rather than e.g. 30 hPa (~24.5 km), from the viewpoints of climate monitoring and process studies (Section 3), satellite validation including radiative transfer calculations (Section 4), and impacts on numerical weather prediction (Section 5). Section 6 provides a summary and concluding remarks.


## 2 Technical issues for balloon soundings

### 2.1 Balloons

The balloons for radiosonde soundings are made of thin rubber. They are filled with either hydrogen or helium gas to obtain

sufficient buoyancy to attain ascent rates, typically 300 m min$^{-1}$ or 5 m s$^{-1}$, which are more or less constant from the surface to the altitude of balloon burst. During ascent, balloons expand with decreasing air pressure, and finally burst when the rubber cannot withstand the tension.  The balloon diameter before burst may become ~5 times that at the surface (see the photo in Bodeker et al., 2016).

The effective burst altitudes are generally determined by various technical and environmental factors. The major technical factors are the balloon size (expressed by balloon mass in grams, e.g. 350 g, 600 g, 1000 g, etc.), the amount of filling gas (see Vömel and Fujiwara, 2021, Section 46.4.4, for typical examples), and the payload. In general, larger balloons will reach higher altitudes for the same payload, but they rarely exceed a pressure altitude of 5 hPa. Extremely large (and expensive) balloons such as 3000 g may reach 40 km. Other technical factors include balloon manufacturer and quality, balloon age, storage

conditions, preparation and inflation procedures, and optional special pre-launch treatments, which will be explained later in this section. Environmental factors include the general atmospheric conditions at a site according to the climatic zone (e.g. special strain on balloons due to extremely low temperatures around the tropical tropopause or the winter polar stratosphere),



time of the day (less cold stress at daytime due to solar heating), and local weather conditions at launch (e.g. wetting by rain or wet clouds with subsequent ice formation during ascent), among others. Note that the systematic nature of the causes of early burst must logically call in to question whether the non-random sampling resulting at the highest heights is truly representative.

Figure 1 shows distributions of the burst altitude cumulative incidence at a resolution of 1 km performed at GRUAN sites between 2005 and 2023. The analysis is based on meta data of more than 130,000 flights with single radiosonde payloads, taken form the GRUAN meta data base (GMDB). The distributions are by balloon size (according to masses from 200 g to 2000 g). No distinction is made for any of the many other factors that can affect the burst height (see below), such as the amount of filling gas or the balloon manufacturer. Overall, Figure 1 confirms the tendency for larger (heavier) balloons to reach higher altitudes, although the height gain tends to decrease with increasing balloon size. However, the analysis presented does not allow quantitative conclusions to be drawn, as the flights included are from only a few measurement sites with their specific operational procedures, sounding conditions, and materials used, in particular the balloon type, all of which may have a specific influence on the statistics of the heights reached.

Recently, in a special research campaign to investigate gravity waves, Kinoshita et al. (2022) successfully made balloon observations up to slightly above 40 km (~3.3 hPa) by using 3000 g balloons and a 40 g radiosonde. This confirms the general assumption that use of larger balloons is the primary solution to reach 10 hPa to 5 hPa. Larger balloons inherently cost more to produce and use more gas, and thus leading to higher costs. Due to the higher expenses for the use of larger balloons and with enough filling gas, networks and stations need scientific justifications (and/or recommendations from e.g. WMO) for a decision in favour of regular observations covering these altitude levels.

Figure 2 illustrates the effects of atmospheric conditions on the cumulative burst point distributions, using data from flights with the same balloon type and size (TOTEX TA600 type 600 g balloons). It can be observed that for both daytime and nighttime, the burst altitudes are generally lower at tropical sites than at higher-latitude sites. The rates of undesired early burst are higher due to the extremely low temperatures (-75°C to -90°C) around the tropical tropopause. The percentage of flights reaching high altitudes is generally larger in daytime than at night, and fewer balloons burst at low and mid altitudes at daytime. This applies more or less in the same way to the three latitudinal regions that are distinguished in the analysis. This is certainly due to the solar radiative heating of the balloon surface, which helps reduce the risk of early burst in the coldest regions in the upper troposphere and lower stratosphere. In particular at tropical sites, early burst rates may also increase due to the higher probability of icing after passing through the very wet lower troposphere. It is interesting to see that burst altitudes at high latitudes are higher than at mid latitudes. This may be related to the special pre-launch balloon treatment practice at some sites which will be discussed in the following, rather than to atmospheric conditions.



There are several simple measures in daily practice that can improve burst heights, essentially without significant additional cost. These include careful handling of the balloons (avoiding contact of the balloon material with other surfaces), keeping the environment clean (e.g. avoiding dust, grit, etc.), wearing gloves, optimising the amount of filling (slightly less filling reduces the rate of ascent slightly, but increases the burst height significantly), filling the balloon as close as possible to the time of launch, and avoiding launches in the rain, especially at night. Some stations may need additional efforts to achieve higher burst altitudes due to challenging climatic conditions. The most challenging factors are the very cold temperatures in the winter polar lower stratosphere and at the night-time tropical tropopause, both without the heating of the rubber balloons by solar radiation. In such cold environments, the rubber may change to a glassy state, resulting in much lower burst altitudes (Vömel and Fujiwara, 2021, Section 46.4.4). To tackle this issue, in addition to using larger balloons, there are three proven methods. The first method is the so-called kerosene treatment, where the balloons are dipped into kerosene or a kerosene-based mixture prior to launch. The second method is to store the balloons in a warm storage (at 55 °C to 60 °C) for hours to days. The second method is often used in combination with the first one. The third method is to use a special balloon system using two balloons, i.e. the so-called double balloon system. An example of such a system is presented by Nash et al. (2011) for summer-time soundings at a southern China station, where a 2000 g balloon is situated inside a 750 g balloon. The outside 750 g balloon protects the inside 2000 g balloon from wetting and low temperatures, assuming that the balloon surface wetting is also a factor of early burst. The 750 g balloon may burst early near the tropopause (~100 hPa) after passing through the wet troposphere and cold tropopause, but after that the fresh 2000g balloon would continue to ascend and may reach e.g. 10 hPa. There is a very recent hypothesis by one of the authors (Kensaku Shimizu) that as the balloon ascends in cold and dry air, friction may result in static electricity accumulating on the balloon outer surface, which may result in spark and premature burst when the balloon rubber becomes quite thin. Actually, 15 experimental flights with 600 g balloons being installed with a "balloon discharger" which is a stick-like material (created with 3D printer, made of plastic) at the balloon neck showed burst altitudes a few kilometers higher (~36 km on average) than those without (~32 km). Further experiments are needed to confirm the effects of this discharger. Also, plastic balloons, which are generally used for much larger payloads than radiosondes, are used at the South Pole for meteorological and ozone soundings (Vömel and Fujiwara, 2021).

Finally, it is noted that automatic radiosonde launchers (ARL) are being used at some operational stations in recent years to reduce personnel expenses (Madonna et al., 2020). Recent ARL models can handle larger balloons (e.g. up to 1000 g). The typical balloon burst heights may not differ much compared to manual launches (see Madonna et al., 2020 for some case studies). But, issues may arise for ARL when surface winds are too strong; this may lead to balloon skin damage at launch, resulting in lower burst altitudes and/or damage to radiosonde sensors at the launch. For manual launches, such risks can be reduced by skilful operations.




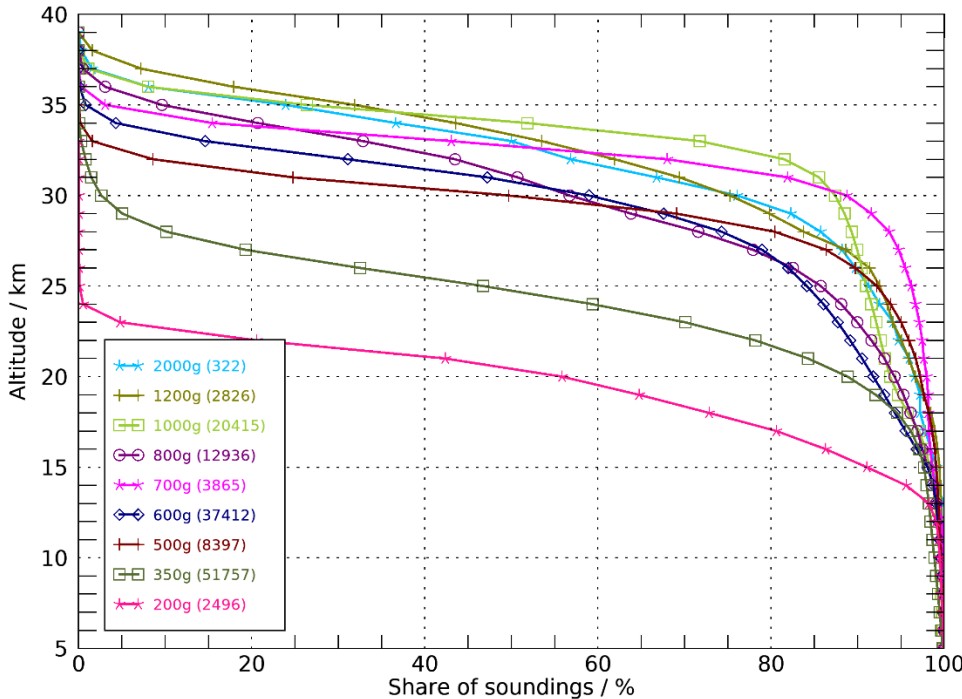


**Figure 1: Cumulative distributions of burst point altitudes at various GRUAN sites (see https://www.gruan.org/network/sites, last access: 25 November 2024) with a single radiosonde as payload and for various balloon sizes from 200 g to 2000 g during 2005–2023.**


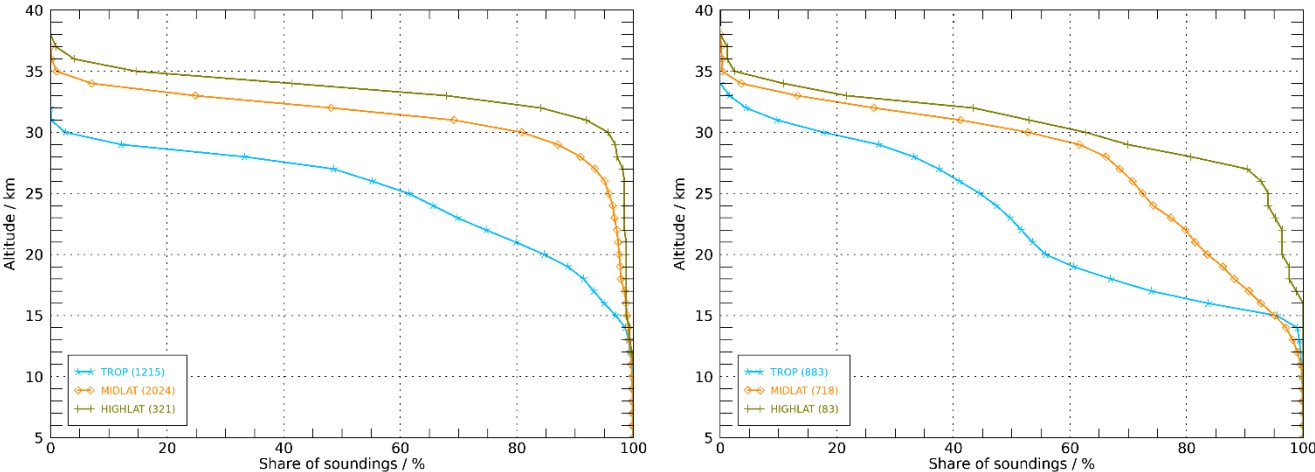



**Figure 2: As for Figure 1 but for the TOTEX TA600 type 600 g balloon at GRUAN sites during 2017–2023, divided by stations located in three latitude regions: between tropical circles (tropical), between tropical and polar circles (mid latitudes), and between polar circles and poles (high latitude). Left panel is for daytime soundings, while right panel is for nighttime soundings.**


### 2.2 Radiosonde sensors

The sensors on radiosondes and special instruments flown together with radiosondes may have their own limitations regarding
the altitude coverage. Here we give a brief review of the recent status of radiosonde sensor technology with a focus on measurement capabilities at high altitudes.

Most of the currently used radiosondes fully utilize the GNSS, in most cases the Global Positioning System (GPS), to measure the position of the radiosonde, and to derive geometric height and horizontal winds (with notable exceptions for Russia and
China where radar tracking systems are still in use at many stations). The uncertainty of the geometric height measurements with such GNSS enabled radiosondes is typically 10 m or less throughout the balloon sounding (e.g. Dirksen et al., 2014; Kizu et al., 2018; Sommer et al., 2023), which is much smaller than that for previous technologies such as ground-based navigation systems and radar systems. The geopotential height is calculated for each data point through the actual ascending trajectory from the measured geometric height and latitude (e.g. Dirksen et al., 2014, Appendix B; Kizu et al., 2018, Section 3.3). An
essential part of the uncertainty for measured horizontal winds comes from the pendulum motions of the payload, which can be reduced by smoothing algorithms at the data processing stage (e.g. Sommer et al., 2023). Therefore, there are virtually no limitations regarding the altitude range for the primary vertical coordinates (geometric and geopotential heights) and horizontal winds measured with "GPS radiosondes". Pressure is generally calculated from the GPS geopotential height measurements using the hydrostatic equation, taking the radiosonde's own measurements of temperature and relative humidity into account,
and is in addition measured with a dedicated pressure sensor in some radiosonde models (e.g. Sommer et al., 2023). It is noted that in general, the uncertainty of pressure measured with a dedicated sensor is roughly constant in pressure (e.g. for Vaisala RS92 radiosonde, 1 hPa for $p > 100$ hPa and 0.6 hPa for $p < 100$ hPa (Dirksen et al., 2014)), while the uncertainty in height measured with a GPS sensor is virtually constant in height. This leads to much greater uncertainty in height measured with a pressure sensor in the stratosphere (e.g. Nash et al., 2011, Section 9; Dirksen et al., 2014, Figure 19), indicating the advantage
of the GPS technology for radiosonde height measurements in the stratosphere.

Solar radiative heating in daytime soundings is the most important error source for the radiosonde temperature measurements (e.g. Dirksen et al., 2014; Kizu et al., 2018; Sommer et al., 2023). The solar radiation heats the surfaces of the sensor and its supporting structures (sensor boom). Due to decreasing efficiency of convective heat exchange with the ambient air with



decreasing pressure, the warm anomalies in temperature readings due to the heating gradually increases with altitude, being most significant in the stratospheric part of a sounding (e.g. Sommer et al., 2023). Beside the actual atmospheric conditions, the radiosonde ventilation, and the solar elevation angle, the strength of the heating is determined by the properties (shape, size, and material) of the sensor construction and its sensitivity to radiation (e.g. Kizu et al., 2018; Sommer et al., 2023). Therefore, the correction algorithms for the solar heating need to be developed separately for each radiosonde model (e.g. Kizu

et al., 2018; Sommer et al., 2023). The overall uncertainty of the radiosonde temperature measurements in the stratosphere at daytime is to a large extent determined by the quality and uncertainty of the radiation correction. Within GRUAN, solar radiation corrections were developed for the Meisei RS-11G and iMS-100 radiosondes (Kizu et al., 2023; Hoshino et al., 2022) and for the Vaisala RS92 and RS41 radiosondes (Dirksen et al., 2014; von Rohden et al., 2022; Sommer et al., 2023). For example, the total uncertainty of temperature measurements after applying the solar radiation correction in the stratosphere

(10 hPa) is 0.3 K to 0.4 K (k=1; where k is the coverage factor corresponding to the level of confidence; see e.g. Immler et al., 2010) for the iMS-100 (Hoshino et al., 2022). The correction algorithm is based on heat balance modelling of the sensor structure and the lead wires, considering absorption of solar energy, heat conduction between the parts of the sensor construction, heat exchange with ambient air, as well as azimuth orientation with respect to the sun and sensor boom angle. Direct solar radiation is modelled by also including a simple parameterization of cloud effects (e.g. Kizu et al., 2018). The

recent GDP for temperature of the RS41 states an overall uncertainty of generally less than 0.2 K (k=1) in the stratospheric part of profiles. Here, solar radiation sensitivity is measured in a specially designed wind tunnel at pressures between the surface pressure value and 5 hPa and at various ventilation rates (von Rohden et al., 2022). The effect of conductive heat exchange with the sensor support structure is inherently taken into account by irradiation of essentially the whole sensor boom. Since the boom orientation relative to the air flow and relative to the incident radiation is important, the sensor boom was

installed during the wind-tunnel measurements at the same angle as in routine soundings, and the radiosonde was constantly rotated to simulate the spinning that occurs during ascents. Profiles of direct and diffuse radiation fluxes were estimated individually for each flight using information from the actual radiosonde measurements and a radiative transfer model based on generic cloud scenarios. Experiments in another extensive laboratory setup at the Korea Research Institute of Standards and Science (Upper Air Simulator, UAS; Lee et al., 2022a) demonstrate that the radiation sensitivity of the RS41 temperature

sensor can be measured with an uncertainty of less than 0.1 K (k=1) at an irradiance of 1360 W m$^{-2}$, considering the effects of the ambient parameters pressure (altitude), absolute temperature, and sensor ventilation.

For reasons of complexity and the required temporal and spatial resolution, real-time cloud information has not yet been used to estimate close-to-reality radiation profiles for actual radiosonde ascents (e.g. Kizu et al., 2018; Sommer et al., 2023). The

uncertainties resulting from the simplifying assumptions or simulations of the radiation situation contribute significantly to the overall uncertainty of the correction. It is noted that due to backscattering (albedo), above cloud layers, the total solar irradiance can exceed the level given by the direct solar irradiance alone by up to 75% (Philipona et al., 2020). There is a multiple-thermometer approach, using multiple sensors each coated with different materials with known radiative property, which may



be able to measure the irradiance and the air temperature at the same time (e.g. Schmidlin et al., 1986; the Lockheed-Martin
Sippican Multithermistor radiosonde explained by Nash et al., 2011). Recently, Lee et al. (2022b) presented results from a
newly developed dual thermistor radiosonde (DTR) which uses two temperature sensors (aluminium and black coated) with
different emissivity. Using the radiation-induced temperature bias between the two sensors, the effective in-situ irradiance is
estimated, based on laboratory-determined radiation sensitivities. The temperature measurement of the working (aluminium
coated) sensor is then corrected based on this irradiance estimate.


The above examples of recent developments in sensor characterization and data products show that modern radiosondes have
the potential to meet current requirements for uncertainties of atmospheric temperature measurements, such as the "Threshold"
and even the "Breakthrough" requirements of 0.5 K and 0.25 K (k=1), respectively, defined by the WMO Observing Systems
Capability Analysis and Review Tool (OSCAR) (https://space.oscar.wmo.int/observingrequirements, last access: 25
November 2024) for the "Atmospheric climate prediction and monitoring" application area, the most demanding area, over
the entire altitude range up to the middle stratosphere. Comprehensive analyses of the measurement performance of ten
different radiosonde models with respect to the OSCAR criteria are presented in the report on the recent WMO Upper-Air
Instrument Intercomparison (UAII2022; Dirksen et al., 2024). Errors due to longwave radiation are of minor importance thanks
to improved sensor coatings (e.g. Kizu et al., 2018; Sommer et al., 2023), and uncertainties of temperature measurements at
night are generally lower than those at daytime. Overall, it can be concluded that in addition to GNSS-derived height, pressure,
and wind, efforts to reach higher altitudes with radiosondes would also be worthwhile with regard to temperature measurements,
regardless of the time of day.

Radiosonde relative humidity (RH) sensors have very limited capability to measure RH in the stratosphere. The main reason
is the overall very low water vapour content at humidities of less than 1 % RH in this region. This is at the lower end of the
calibration range of thin-film polymer capacitive sensors for all modern radiosondes where the measurement uncertainties
exceed the measured values. It has been shown, however, that radiosondes are still able to detect exceptional events such as
the Hunga Tonga–Hunga Haʻapai volcanic eruption in January 2022 (Vömel et al., 2022; see Section 3), where large amounts
of water vapour were injected into the stratosphere and distributed on a global scale. Large response times due to low
temperatures are another limiting factor for thin-film polymer capacitive sensors (e.g. Kizu et al., 2018; Sommer et al., 2023),
in particular at very low temperatures around the tropopause. Special types of instruments are used for measuring stratospheric
water vapor using balloons as described below.

There are a number of special instruments that are combined with radiosondes and regularly launched on the same balloon.
The radiosondes there act as transmitters of data measured with the special instruments, and provide important data e.g. for
the vertical coordinate and conversion of the raw values to concentrations. Examples include ozonesondes and frost-point
hygrometers. In the global ozonesonde network, most stations use the electrochemical concentration cell (ECC) sensor. WMO





(2021b) fully discusses the uncertainty budget of the ozonesonde measurements and shows that the total uncertainty in ozone partial pressures is less than 5 % to 10 % below 30 km (except for the tropical tropopause region) but becomes much greater

above 30 km where further technological developments are needed. Frost-point hygrometers are used as instruments for balloon-borne in-situ measurements of water vapour, especially designed for accurate measurements of low stratospheric concentrations. The Cryogenic Frostpoint Hygrometer (CFH; Vömel et al., 2007, 2016) and National Oceanic and Atmospheric Administration (NOAA) Frost Point Hygrometer (FPH; Hall et al., 2016; Hurst et al., 2023), both using cryogen materials for mirror cooling, are established and widely used instruments for research, satellite validation, and monitoring (e.g. SPARC,

2000; Kiefer et al., 2023). The total uncertainty of water vapor mixing ratios in the stratosphere is typically evaluated as 2 % to 3 % up to ~25 km. Other frost-point instruments using different mirror cooling techniques include Meteolabor Snow White (Fujiwara et al., 2003; see its limitations in Vömel et al., 2003), Meisei SKYDEW (Sugidachi et al., 2024), and PCFH (Brossi, 2024). Outgassing of water vapor from the balloon, parachute, and instrument package, however, may lead to measurement contaminations above ~25 km, which are currently identified visually (Vömel et al., 2016). At some sites, a controlled balloon

descent technique has been used to avoid contaminations from the balloon wake, with a starting point of descent around 29 km (Kräuchi et al., 2016). This means that further technological investigation is needed for frost-point-hygrometer sounding systems above ~29 km. There are also some balloon-borne backscatter instruments for particle measurements flying in combination with radiosondes (e.g. Suortti et al., 2001; Brunamonti et al., 2018), and balloon-borne systems measuring upwelling and downwelling radiation profiles (e.g. Philipona et al., 2012). These instruments have no fundamental limitations

up to ~10 hPa level.

In summary, modern GPS radiosondes can measure height and horizontal winds over the entire balloon profile. Solar radiative heating in daytime soundings is the most important error source for the radiosonde temperature measurements. Modern radiosondes, however, have the potential to meet e.g. current WMO OSCAR requirements for uncertainties of temperature

measurements, thanks to recent developments in sensor characterization and data products, in particular by GRUAN.

## 3 Climate monitoring and atmospheric process studies

In this section, the importance of upper-air measurements for the height region 30 hPa to 5 hPa (24.5 km to 37 km) is discussed for both climate studies and atmospheric process studies.

Long-term cooling trends have been observed in the stratosphere at least for the past 40 years, which are primarily due to the increase of greenhouse gases with modulations by evolving ozone changes (e.g. Steiner et al., 2020b and the references therein;

Santer et al., 2023). As explained in Sections 1 and 2, several homogenized radiosonde data sets are usually used in trend



studies (e.g. Steiner et al., 2020b; Zhou et al., 2021; Madonna et al., 2022), but technological improvements in modern radiosonde models and efforts such as those by the GRUAN (Bodeker et al., 2016) will result in providing upper air data that can directly be used for climate studies in the near future without homogenisation. We do need long-term monitoring of the full atmospheric column, and the height region of 30 hPa to5  hPa is highly sensitive to changes in green-house gases and the ozone layer (e.g. Hufnagl et al., 2023). For instance, recent work has demonstrated that the global-mean radiative forcing for a change in $CO_2$ concentration is sensitive to the background temperatures at ~10 hPa, which is the effective emission level for the centre of the $CO_2$ absorption band (Jeevanjee et al., 2021; He et al., 2023). Consequently, substantial uncertainty in temperatures at this level across climate models accounts for roughly half of the inter-model spread in $CO_2$ radiative forcing (He et al., 2023). This model spread has persisted for decades (Soden et al. 2018), and its dependence on stratospheric temperatures suggests that it could be constrained by consistent high quality temperature observations of the stratosphere. Continuous observations will be required because evolving temperatures at 10 hPa has led to, and will continue to lead to changes in the magnitude of $CO_2$ radiative forcing with time for a given concentration change (He et al. 2023). Historical sampling at the highest heights may systematically only sample certain states because the causes of early balloon burst are principally systematic and in many cases related to cold tropopause region temperatures (Section 2).

As a reference network, GRUAN also provides a potential basis for enhanced interpretation of broader radiosonde networks, e.g. through the provision of instrumental corrections which can be applied to data from non-GRUAN stations to adjust quantifiable systematic effects compromising the quality of operationally processed data. In the past, efforts were spent to assess the impact of the GRUAN radiation corrections on historical radiosonde data (Wang et al., 2013). More recently, taking advantage of the GRUAN and WMO radiosonde intercomparison data, a novel approach called the Radiosounding HARMonization (RHARM) for homogenizing historical radiosounding data since 1978 has been provided (Madonna et al., 2022). The RHARM is a hybrid method, which provides an adjustment of IGRA radiosounding observations (Durre et al., 2018) of temperature, humidity, and wind from 2004 to present using the GRUAN data and algorithms, as well as the 2010 WMO/CIMO radiosonde intercomparison dataset (Nash et al., 2011), combined with a quantification of measurement uncertainties. The benefit of this GRUAN-based approach is shown in Figure 3, where the IGRA and the RHARM temperatures at 10 hPa have been compared with the RS92 GRUAN data product version 2 (RS92-GDP.2, Dirksen et al., 2014). The comparison includes 00 and 12 UTC ascents at 8 GRUAN stations (selected on the basis of the data record length or density) from 2008 to 2018. Fig. 3 shows that the RHARM approach can reduce the bias, on average, in the operationally processed IGRA data also at 10 hPa level, both in daytime mimicking the GRUAN radiation correction and at night where a statistical adjustment is obtained from the comparison with the GRUAN GDP. These results are an example of how reference-quality upper-air data may positively impact the more spatially extensive non-reference soundings data in the mid and upper stratosphere, contributing to better characterization of climate change and enhancing satellite validation.



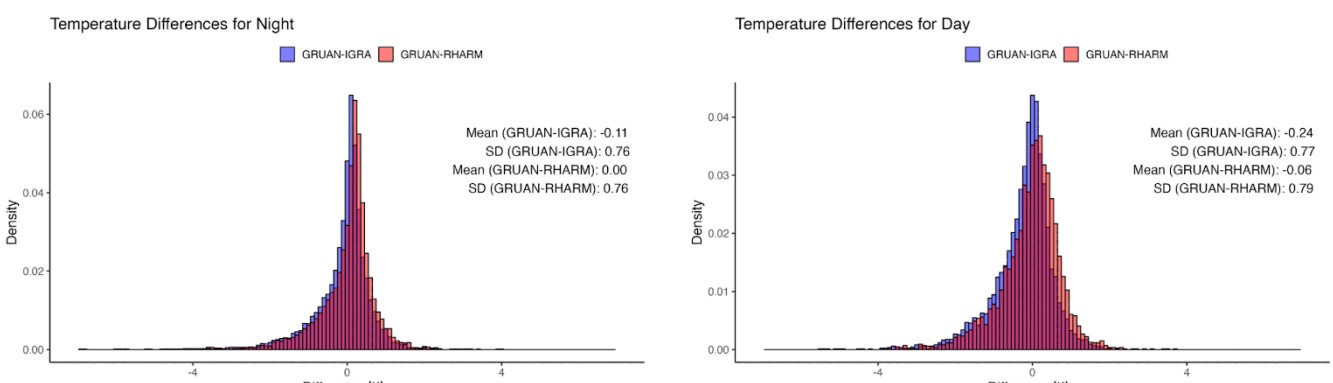

**Figure 3: Histogram of night-time (left) and daytime (right) temperature differences between IGRA and GRUAN (blue) and between RHARM and GRUAN (red) at 10 hPa at 8 GRUAN stations (Cabauw, Lamont, Lauder, Lindenberg, Ny-Ålesund, Payerne, Sodankyla, and Tenerife) where RS92 GRUAN data product version 2 is available for the period from 2008 to 2018. All the available ascents have been considered. Note that the starting and ending dates are different at different stations, depending on the sonde type transition date at each station. For example, in 2008, there were only two stations, Lindenberg and Ny-Ålesund. The high-resolution**
**GRUAN RS92 data (at 10 m) were matched with IGRA and RHARM data, that are available only at mandatory and significant levels through a linear interpolation, of levels using the two nearest levels, respectively, above and below 10 hPa. The bias between GRUAN data and RHARM data as well as the random uncertainty for both data sets are discussed in Madonna et al. (2022).**

Among surface-based measurement techniques, radiosondes and Raman/Rayleigh lidars are often used in synergy or in a redundant way to investigate atmospheric temperature and water vapour (e.g. Whiteman et al., 2006; Dabas et al., 2008). Radiosondes are also one of the very few sources of horizontal wind measurements in the stratosphere at the global scale and with a high vertical resolution. Doppler radars and lidars can measure winds, but typically only up to the upper troposphere (see e.g. Foken, 2021). There are a few so-called mesosphere-stratosphere-troposphere (MST) radars around the world for
studies on atmospheric dynamics, but they are not sensitive in the region around 25 km to 60 km (e.g. Hocking et al., 2016; Sato et al., 2023). There are also some Doppler lidars that can provide vertical wind profiles in the lower stratosphere, up to 30 km, detecting the Rayleigh–Brillouin scattering (Tenti et al., 1974), whose vertical resolution is typically of a few hundred metres. Therefore, radiosonde wind measurements, provided at e.g. 20 sec (~100 m) resolution, still represent the highest resolution information on winds in the stratosphere. Note that regarding satellite measurements, between August 2018 and
April 2023, the Atmospheric Dynamics Mission-Aeolus (ADM-Aeolus), of the European Space Agency (ESA) has been the first satellite with equipment capable of performing global wind-component-profile observation from the Earth's surface into the stratosphere (Flamant et al., 2020; Rennie et al., 2021; see also AMT/ACP/WCD inter-journal special issue on "Aeolus data and their application" at https://amt.copernicus.org/articles/special_issue1131.html, last access: 25 November 2024; https://earth.esa.int/eogateway/news/the-divine-keeper-of-the-winds-retires, last access: 25 November 2024).




Other than these direct measurements, horizontal winds in the extratropical stratosphere are derived from temperature measurements from satellites, radiosondes, and aircraft through the thermal wind relationship, e.g. in the data assimilation procedure within numerical analysis/weather-forecast systems. The thermal wind relationship is a very good approximation for synoptic to planetary scale motions at middle and high latitudes, in particular in winter. In the tropical stratosphere, zonal

wind shows a very unique oscillation with a period slightly greater than 2 years, i.e. the Quasi-Biennial Oscillation which spans from ~16 km (~100 hPa) to 40 km (~3 hPa) (Baldwin et al., 2001, 2019; Hitchman et al., 2021; Haynes et al., 2021; Anstey et al., 2022). The QBO is known to be driven by various types of equatorial waves that are generated by organized tropical convection and propagating and dissipating through the stratosphere. Because of the breakdown of the thermal wind relationship in the tropics, the QBO has been monitored primarily with radiosonde zonal wind data taken at Singapore (see e.g.

Fujiwara et al., 2020) and a few other equatorial stations (see e.g. https://www.atmohub.kit.edu/english/807.php, last access: 25 November 2024). Tropical radiosonde wind data are very important to constrain tropical stratospheric zonal winds in global atmospheric reanalysis systems (Kawatani et al., 2016; Essa et al., 2023). Anstey et al. (2022) summarized the role of the QBO in global atmosphere and in climate: The QBO has teleconnections to phenomena outside the tropical stratosphere and affects seasonal predictability globally; monitoring of the QBO amplitude is important for climate monitoring because climate models

project its future weakening; and although the QBO phase changes have historically been very predictable, since 2016 its regular cycling has been disrupted twice, for reasons not yet well understood. All these strongly suggest the importance of tropical radiosonde wind measurements covering daily to interannual time scales in the 30 hPa to 5 hPa region.

In the polar stratosphere, Sudden Stratospheric Warmings (SSWs) are a dramatic phenomenon in winter, which are

characterized by warming of as much as 30 K to 50 K within a couple of days together with abrupt deceleration of the climatological westerly circulation of the polar vortex, in association with planetary wave activity (e.g. Baldwin et al., 2021). There are several remote effects arising from SSWs on the atmosphere both above and below the stratosphere, including surface weather and its predictability (Baldwin et al., 2021; Nie et al., 2019; Scaife et al., 2016; Kidston et al., 2015). For example, given an adequate observational constraint, numerical weather prediction models with high model tops are typically

able to predict the onset of SSWs more than 5 days prior (Tripathi et al., 2015), although significant event-to-event variability in predictability has been demonstrated (Karpechko et al., 2018). Also, less wave activity and fewer and weaker SSWs lead to a stronger and colder winter stratospheric polar vortex, resulting in more polar stratospheric clouds (PSCs) resulting in chemical ozone depletion and modulating tropospheric circulation regimes (e.g. Manney et al., 2022 and the references therein). Therefore, the full characterization of the state of winter polar stratospheric temperature and winds at daily to interannual time

scales is very important for sub-seasonal to seasonal prediction, climate studies, and ozone layer studies. Reference-quality and high-vertical-resolution radiosonde temperature and wind data, measured simultaneously and covering the 30 hPa to 5 hPa region, would provide a more comprehensive understanding of atmospheric dynamics, and are potentially unique data for evaluation of weather-prediction, climate models, and reanalysis data sets.



Figure 4 shows times series of temperature and zonal wind at 10 hPa over Ny-Ålesund (78.92°N, 11.93°E) in the Arctic Ocean from December 2015 to March 2017 using GRUAN and IGRA radiosonde data products and two global atmospheric reanalysis data sets, ERA5 (Hersbach et al., 2020) and JRA-3Q (Kosaka et al., 2024). In general, the four data sets show good agreement during the boreal summer for both temperature and zonal wind. During the two boreal winters, a few SSW events can be identified (e.g. Eswaraiah et al., 2017; Mitnik et al., 2018). Fig. 4 shows that the reanalyses capture the warming spikes during

January-February 2016, although for the second SSW in February 2016 the highest temperature value in the reanalyses is ~8 K higher than that from the two upper-air data products. During January-March 2017, on the other hand, the number of radiosondes that reached 10 hPa is less, and thus it is not possible to fully evaluate the reanalysis performance during these SSWs. ERA5 and JRA-3Q time series are characterized by a difference over the entire time series within about 1 K with the largest difference during the SSW temperature peaks. For zonal wind, the comparison reveals much greater differences among

the data sets, and even between the two radiosonde data products during the boreal winters in association with SSWs. The differences are likely due to the different vertical resolutions of the three datasets and to the related capability to properly measure the ageostrophic wind component that may generate differences larger than 5 m s$^{-1}$ in the stratosphere (Nimac et al., 2024). Increased wave activity related to SSWs mainly contributes to producing ageostrophic effects (Elson, 1986). We note again that during January-February 2017, a smaller number of radiosonde ascents reached 10 hPa due to premature balloon

bursts (6 out of 31 ascents in January and 6 out of 28 in February), which prohibits a full evaluation of the reanalyses. Previous work on the GRUAN's representativeness (Weatherhead et al., 2017) revealed that for stratospheric to upper-tropospheric temperature, approximately 8 % to 10 % of the Earth is well correlated with at least one GRUAN station. However, the stations in the European sub-polar and polar regions (i.e. Ny-Ålesund and Sodankylä) have much larger correlation areas, potentially because the orographic features are minimal in the same regions and atmospheric dynamics allow for larger spatial coherence.

However, the comparison between the radiosonde and reanalyses data is also affected by the representativeness error, because of choosing the nearest grid point from the reanalysis data to the station coordinates (Madonna et al., 2023). A comparison with the nearest grid point to the exact position of the flying balloon might further minimize the differences, especially in atmospheric events with a large spatial variability.






**Figure 4: Time series of temperature (top) and zonal wind (bottom) at 10 hPa over the Ny-Ålesund station (78.92 °N, 11.93 °E, 5 m asl) using GRUAN (black circles) and IGRA (red circles) radiosonde data products, and ERA5 (blue lines) and JRA-3Q (green lines) reanalyses. For the two reanalyses, data at the nearest grid point to the station coordinates have been used.**

The strength of radiosonde measurements is that signatures of atmospheric gravity waves in temperature and winds are simultaneously captured (e.g. Tsuda et al., 1994; Geller et al., 2013; Okui and Sato, 2020) as is the case for the SSWs study shown in Figure 4. Atmospheric gravity waves are ubiquitous, intermittent sub-synoptic scale waves generated by the orography and non-orographic processes including convective and jet-front systems, propagating both horizontally and vertically and dissipating in the stratosphere and mesosphere (e.g. Baldwin et al., 2019, Section 6; Fritts and Alexander, 2003).



They redistribute momentum and energy in the atmosphere through their generation, propagation, and dissipation, being the key driving factor for the middle atmosphere circulations including the QBO, the deep part of Brewer-Dobson circulation, and the mesospheric meridional circulation. Therefore, their effects need to be considered in numerical weather forecast models including the reanalysis systems as well as climate models, but because of their small spatial and temporal scales, parameterisations are needed for sub-grid-scale gravity waves even in recent high-resolution models (e.g. Hersbach et al., 2020). Gravity waves have been observed with the MST radars, balloons, and rockets, and more recently also with high-resolution satellite measurements (e.g. Kalisch and Chun, 2021; see also Baldwin et al., 2019). Recently, by using a very large 3000 g balloon, radiosondes were flown up to ~40 km for gravity wave investigations (Dörnbrack et al., 2018; Kinoshita et al., 2022) because radiosonde soundings are still a key tool to investigate gravity waves in the height region from ~25 km to ~40 km. Radiosonde data products such as the GRUAN data products, with high vertical resolution (i.e. 1 sec, or ~5 m) and with appropriate data processing to account for solar radiation correction for temperature and pendulum motion removal for winds, would be very useful, if they consistently reach 10 hPa to 5 hPa levels, for further investigation of gravity waves.

It is possible that the climate system may enter into a new phase of global warming (e.g. tipping points) with abrupt and even irreversible changes in the system (IPCC, 2021). From the viewpoint of our observing systems, we may need to prepare for "surprising" phenomena, being previously unknown and/or not well recognized. One recent example for a previously unknown large-scale event is the January 2022 eruption of Hunga Tonga-Hunga Ha'apai submarine volcano, which injected unprecedented amounts of water vapour into the stratosphere (Vömel et al., 2022). Given the tremendous size of the signal, Vömel et al. (2022) found that Vaisala RS41 radiosonde RH data, after careful reprocessing, could be used to measure the water vapour mixing ratios of the volcanic layers in the stratosphere up to the balloon ceiling altitudes (e.g. ~28 km to ~31 km, depending on each station). Many operational stations in the Southern Hemisphere were using Vaisala RS41 and flying it up to ~30 km at that time, providing very valuable and unique information on the evolution and transport of the volcanic water vapour layers during the first three months. Large-scale climate geoengineering/intervention proposals including the solar radiation modification to slow the warming at least temporarily are more seriously discussed in recent years (e.g. IPCC, 2021; Visioni et al., 2023; see also https://csl.noaa.gov/research/erb/, last access: 25 November 2024); if some of them are to be tested in the atmosphere or even implemented in the future, high altitude operational radiosoundings both before and after such tests would be very important to evaluate the effectiveness, influences, and side effects.

## 4 Satellite CAL/VAL

Space programmes dedicated to Earth observation are extremely expensive, often of the order of a billion Euros. Although initial costs are high, the socio-economic benefit is far larger. For example, the EUMETSAT Polar System-Second Generation



(EPS-SG) programme at a cost of €3.4 billion is expected to yield a benefit to cost ratio ranging from 5 to 20 (EUMETSAT, 2014). To ensure a maximised return on investment, data quality must be delivered with the highest possible standard, which requires, among other steps, performing a thorough CAL/VAL with quantified SI traceable uncertainty where possible.

There is not, however, a one-size-fits-all CAL/VAL solution, nor a single comprehensive method, but rather an array of complementary strategies and techniques that encompass reliable calibration, intercalibration via simultaneous overpasses and/or double difference through a transfer reference, and field campaigns making use of ground-based and airborne in situ and remote instruments (e.g. Cao et al., 2004; Larar et al., 2010; Wang et al., 2011; Müller, 2014, Cimini et al., 2024).

Space agencies also collaborate with NWP centres to assess instrument performance against their NWP models. The comparison of satellite observations with NWP models presents several advantages compared to other CAL/VAL methods: there is no sampling gap; it is continuously available from the surface to the top of the atmosphere; the physics is constrained so that geophysical fields remain consistent; an optimal state of the atmosphere is obtained from state-of-the-art forecast models and data assimilation systems; and performances are continuously monitored against independent reference observations and 555 analyses (Saunders et al., 2013; Newman et al., 2020).

There are several satellite dedicated CAL/VAL initiatives including of direct relevance to radiosonde measurements: (1) Under the WMO, the Global Space-based Inter-Calibration System (GSICS, https://gsics.wmo.int/, last access: 25 November 2024) is promoting international collaborations for operational environmental satellites for climate monitoring and weather 560 forecasting; and (2) The NOAA Products Validation System (NPROVS, https://www.star.nesdis.noaa.gov/smcd/opdb/nprovs/, last access: 25 November 2024) routinely monitors collocated atmospheric data, by compiling collocated data from radiosondes, dropsondes, numerical model outputs, and various satellite observations (Sun et al., 2023). For the purpose of direct satellite CAL/VAL with radiosondes, collocation (i.e. overpass) and the highest possible sounding ceiling are the most important aspects for radiosonde data to be able to be fully utilized (e.g. Calbet et al., 2011; Carminati et al., 2019; Vömel and 565 Ingleby, 2023).





A baseline summary of currently available high-ascent WMO radiosondes reaching at least 10 hPa with an emphasis on those reaching 5 hPa (37 km) highlights present issues. Among approximately 18,000 WMO operational conventional radiosonde sounding data sampled during a 15-day period in March 2024, approximately one-third reached 10 hPa (32 km) with about 4% attaining 5 hPa (37 km).  Among sites launching Vaisala RS41 radiosondes (analysed later in this report), currently comprising about 40 % of the WMO network, 53 % (3626 soundings) reached 10 hPa and about 6 % (433 soundings) attained 5 hPa.  High ascent Vaisala RS41 reports reaching 5 hPa are considered most valuable to anchor satellite stratospheric CAL/VAL (radiometric and geophysical); their global distribution is shown in the upper panel of Figure 5 (dots).  Also shown are GRUAN sites (indicated as "G") for an overlapping period from April 2023 through May 2024 for which at least one radiosonde that reached 5 hPa was observed, about 1% (88 soundings) of the 9347 GRUAN RS41 sampled; approximately one-third (3276 soundings) reached 10 hPa.  As can be seen, the global distribution of conventional RS41 radiosondes reaching 5 hPa is reasonably robust latitudinally, but longitudinally skewed with over 75 % of the reports in North America and Europe. The distribution of GRUAN reports reaching 5 hPa is limited to the vicinity of Europe and Japan with Payerne (Switzerland) leading the way with almost 10 % of launches reaching 5 hPa.

The subsets of conventional high ascent RS41 radiosondes reaching 5 hPa that are "naturally" (or operationally) collocated within 1-hour (in the stratosphere) of EUMETSAT and NOAA polar satellites, respectively, are shown in the middle and bottom panels of Fig. 5. These currently have the highest value for satellite based stratospheric CAL/VAL.  High ascent radiosondes that are naturally collocated with GNSS RO (not shown) are rare, but later in Section 5.3, the global distributions of radiosondes, ECMWF analysis and GNSS RO (COSMIC2 and GRAS, respectively) all collocated within 2-hours are analysed.

The frequency of operational high ascent radiosondes collocated with specific satellites during the 15-day March 2024 period (Fig. 5) were: 39 for NOAA; 36 for MetOp; 1 for COSMIC2; and 0 for MetOp GRAS. There were no operational ascent "golden" collocations of radiosondes reaching 5 hPa with both polar and GNSS RO observations all within 1-hour in the stratosphere.

Increasing the frequency and global coverage of radiosondes reaching 5 hPa that target polar and GNSS RO satellite overpass, and as feasible "golden collocations targeting both" is encouraged. This could be optimally achieved by leveraging existing NOAA and pending EUMETSAT targeted (dedicated) radiosonde programs. Their direct utilization in satellite data CAL/VAL and feedback to radiosonde technology, weather, and climate communities would be expeditious. Targeting high ascent GRUAN radiosondes would be optimal.









**Figure 5:** The upper panel shows the global distribution of conventional RS41 radiosondes (dots) that reached 5 hPa during a 15-day period, March 2–18 2024, and the GRUAN sites (indicated as "G") that reached 5 hPa for the period April 2023 through May 2024. The lower 2 panels show subsets of conventional RS41 reaching 5 hPa (upper panel) that were operationally collocated within 1-hour of NOAA (middle) and MetOp (bottom) satellite overpass in the stratosphere. Note that it takes approximately 1 hour for the radiosonde to reach 100 hPa and another hour to reach 5 hPa.

## 4.1 Radiative transfer calculations for microwave and infrared sounders

Fast radiative transfer (RT) models are used to calculate top-of-atmosphere simulated radiance from NWP fields which are then used for comparison with satellite data. The unique solution obtained through forward RT calculation is an advantage over the multi-solution ambiguity that arises from inverse methods (i.e., retrievals). Newman et al. (2020) summarised the NWP-based CAL/VAL work conducted for several satellite instruments during gap analysis for the integrated atmospheric ECV climate monitoring (GAIA-CLIM) project. The authors stress, however, that the uncertainties in the NWP systems are not (yet) traceable but that this gap can be addressed to some extent by linking NWP fields to traceable reference observations such as GRUAN radiosondes. The GRUAN Processor, also developed during GAIA-CLIM in response to this identified gap (Carminati et al., 2019), provides the infrastructure to collocate GRUAN and NWP profiles and to simulate top-of-atmosphere radiance for all instruments supported by the underlying radiative transfer model developed and distributed by the Satellite Application Facility for Numerical Weather Prediction (NWP SAF), RTTOV (Saunders et al., 2018). The processor enables the SI traceable uncertainty from GRUAN profiles to be propagated into radiance-space allowing a robust statistical analysis of the comparison between GRUAN and the NWP model under assessment.

Newman et al. (2020) used the GRUAN Processor to evaluate the potential for NWP models to be used for CAL/VAL of EUMETSAT future EPS-SG mission, with a focus on the Infrared Atmospheric Sounding Interferometer-Next Generation (IASI-NG) and Microwave Sounder (MWS) instruments (Holmlund et al., 2017; Cimini et al., 2023). Based on a 6-month sample of RS41 GDP, the authors were able to demonstrate that for simulated temperature sounding channels, the NWP and GRUAN data are metrologically consistent with a total combined uncertainty ranging from 0.1 K to 0.4 K. When assessing future IASI-NG observations, great care needs to be taken to account for additional sources of uncertainties. These may arise from the direct comparison between the NWP and satellite radiances since the total uncertainty for a given channel should not exceed the instrumental radiometric accuracy requirements of 0.25 K for a meaningful detection of biases of that order. For the MWS, a metrologically robust NWP-based assessment will be possible given the instrumental radiometric accuracy requirement of 1 K. On the other hand, the total uncertainty between NWP models and GRUAN is in the 1–3 K range for



simulated humidity sounding channels. This means that the NWP CAL/VAL will be viable only for biases of that order or larger.

The Newman et al. (2020) study was however limited by the radiosonde ceilings as only channels with Jacobian peaking below radiosonde profile top were investigated. As a result, MWS channels 13, 14, 15 and 16 (near the 57.290344 GHz $O_2$ absorption

line, see e.g. https://space.oscar.wmo.int/instruments/view/mws, last access: 25 November 2024), respectively peaking at 30, 15, 6, and 3 hPa (similarly to ATMS channels in Figure 7) were ignored. Furthermore, IASI-NG channels with wavenumbers less than 657.5 cm$^{-1}$ were also ignored. For example, IASI-NG will have 84 channels peaking between 30 and 5 hPa, with wavelengths ranging from 645.875 to 720.75 cm$^{-1}$. The sample size above 10 hPa drops sharply (see, e.g., Figure 4b of Carminati et al., 2019) making it more difficult to perform a metrologically rigorous comparison particularly as ascents

reaching that height may not be representative.

Radiosonde profiles are also directly fed into RT models to compare top-of-atmosphere simulated radiances with the corresponding microwave (MW) and infrared (IR) observations (Berg et al., 2016; Brogniez et al., 2016, Calbet et al., 2011, 2017, 2018, 2022; Sun et al., 2021; Cimini et al., 2024). To quantify the impact of the ceiling heights of radiosonde profiles

on satellite validation, the following simulated experiment was performed using measurements from a set of 83 high-altitude radiosondes launched from five GRUAN sites, i.e. Neumayer (Antarctica), Ny-Ålesund (Svalbard), Lindenberg (Germany), Payerne (Switzerland), and Potenza (Italy). Using these radiosondes, two runs of RT computations were performed: the first with the original radiosonde ceiling, and the second with the profiles truncated at pressures below 30 hPa (i.e. data at altitudes higher than the 30-hPa level were removed), thus mimicking low-altitude ceilings. In both cases, profiles are extended above

the ceiling with a climatological mean and then with the nearest ERA5 reanalysis estimate. To highlight the impact of the truncated radiosonde ceiling, profile differences are shown in Figure 6 for temperature and relative humidity for both types of profile extensions, climatology and ERA5. Note that the profiles topped with climatology result in large temperature differences above 30 hPa (left), on the order of -30K to +10K, while the ERA topped profile differences are on the order of ±5K. The original and modified profiles are then used to compute top-of-atmosphere brightness temperatures ($T_B$) and the

differences between the simulations computed in the two runs ($\Delta T_B$) are taken as the additional error caused by a limited radiosonde profile ceiling. The standard deviation of $\Delta T_B$, over the whole dataset of 83 profiles ($std(\Delta T_B)$), is taken as an indication of the additional random uncertainty (1-sigma) caused by truncating the profiles. This analysis was performed both in the MW and IR spectral regions to model the impact of radiosonde ceilings on the remote sounding instruments including the Advanced Technology Microwave Sounder (ATMS; Zou et al., 2018) and the Infrared Atmospheric Sounding

Interferometer (IASI; August et al., 2017). This analysis would apply similarly to future MWS and IASI-NG instruments.





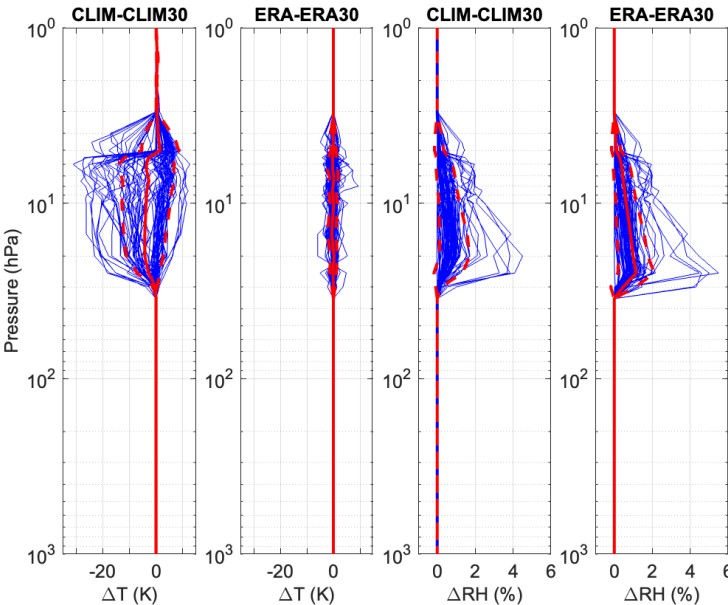

**Figure 6: Temperature (T; left and 2nd subplot) and relative humidity (RH; 3rd subplot and right) differences for RT input profiles extended with climatology (CLIM) and ERA5 (ERA) (see legends above each panel). Differences shown (in blue) are with the high-altitude profiles minus the low-altitude profile (truncated at 30 hPa). The mean and mean ±1 standard deviation are shown in solid-red and dashed-red respectively.**

### 4.1.1 Results for microwave wavelengths

The MW calculations were performed using PyRTlib (Larosa et al., 2024), a line-by-line RT code based on Rosenkranz atmospheric absorption model (Rosenkranz, 1998) and later modifications (Rosenkranz, 2017; Rosenkranz and Cimini, 2019; Gallucci et al., 2024). The sensitivity of selected ATMS channels to atmospheric temperature is shown in Figure 7 (adapted from Zou et al., 2018). The additional random uncertainty for ATMS temperature-sensitive channels is shown in Figure 8 by using profiles that are extended vertically beyond their artificial cut-off with the climatology and ERA5 data. As one may have expected, the additional uncertainty is low for channels sensitive to tropospheric temperatures (ATMS channels 3–5), while it increases for channels sensitive to stratospheric temperatures (e.g., ATMS channels 10–15). In particular, the additional uncertainty remains below 0.5 K if the profiles are topped with ERA5 but reaches nearly 6 K if the profiles are topped with a climatological mean. This clearly demonstrates the need to extend radiosonde profiles above their ceiling using reanalysis data, and it also shows the importance of high radiosonde ceiling altitudes for the validation of temperature sounding channels of





ATMS and more generally of any satellite sensor channels sensitive to these altitude regions. This is particularly so given the

risk of a degree of circularity in using ERA5 if it assimilates the target radiances being validated.

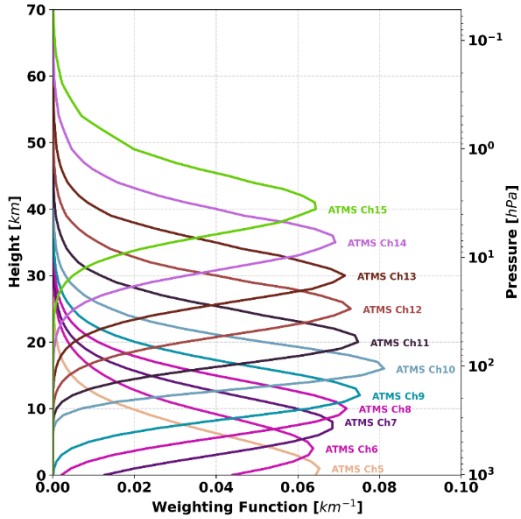

**Figure 7: Weighting functions for temperature-sensitive ATMS channels 5 to 15, computed at 1km resolution (adapted from Zou et al., 2018). These show that channel 13 has the largest sensitivity in the region of interest, peaking between 10-20 hPa. Weighting functions of channels 14 and 15 peak higher in the atmosphere, indicating lower sensitivity to the region below 10 hPa.**

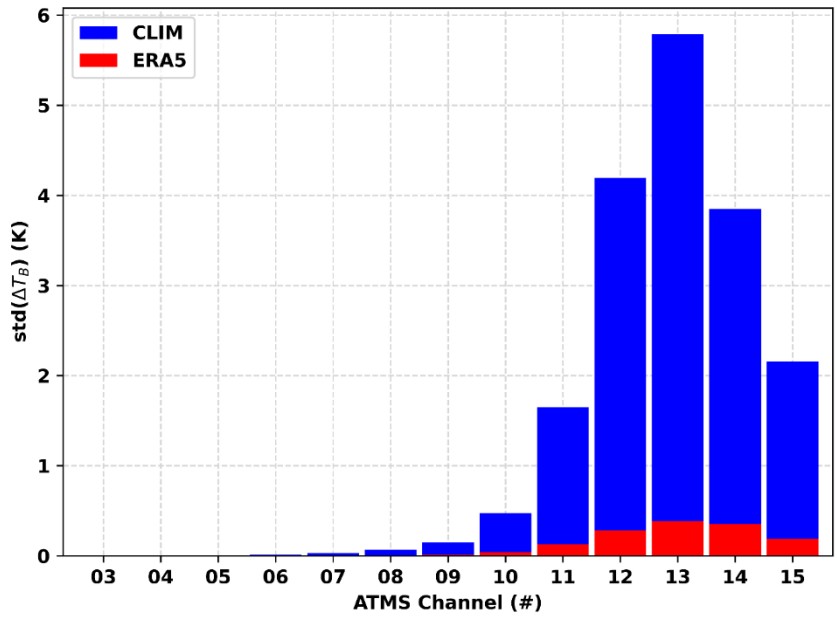



**Figure 8: Additional random uncertainty ($std(\Delta T_B)$) for the temperature-sensitive channels of the Advanced Technology Microwave Sounder (ATMS). Blue (red) bars indicate additional uncertainty for radiosonde profiles topped with climatological means (nearest ERA5). Channel frequency (in GHz) is respectively (number after ± indicate the displacement of spectral side bands with respect to central frequency): 50.3 (Ch3), 51.76 (Ch4), 52.8 (Ch5), 53.596 ± 0.115 (Ch6), 54.4 (Ch7), 54.94 (Ch8), 55.5 (Ch9), 57.29 (Ch10), 57.290 ± 0.217 (Ch11), 57.290 ± 0.322 ± 0.048 (Ch12), 57.290 ± 0.322 ± 0.022 (Ch13), 57.290 ± 0.322 ± 0.010 (Ch14), 57.290 ± 0.322 ±**
**0.0045 (Ch15). See text for the details of the numerical experiment.**

### 4.1.2 Results for infrared wavelengths

The IR calculations were performed using the Atmospheric and Environmental Research Line-by-Line Radiative Transfer Model (LBLRTM) double precision version 12.6 (Clough et al., 2005). This model uses the 'aer_v_3.5' line parameter
database and is based on the HITRAN 2012 database (Rothman et al., 2013) with updates made to the $CO_2$, $H_2O$, $CH_4$, and $O_2$ line parameters (Benner et al., 2016; Devi et al., 2016; Drouin et al., 2017; Oyafuso et al., 2017). The IR analysis was performed using the same methodology and input profiles that were used in the MW region, where the 83 temperature and relative humidity profiles from radiosondes were used up to the original high-altitude ceilings and then again truncated at 30-hPa, to model low-altitude ceilings. In addition, these high and low profiles were extended using either a climatological mean or the
nearest ERA5 reanalysis to 1 hPa. These four sets of profiles were used to compute top-of-atmosphere IR brightness temperatures ($T_B$) using LBLRTM and conditioned to the IASI spectral region and resolution.

The standard deviation of the brightness temperature differences ($std(\Delta T_B)$), an indication of the additional random uncertainty due to the truncated radiosonde ceiling, is shown in Figure 9 for each type of profile extension. The additional
uncertainty is largest for the climatology (CLIM) topped profiles in four spectral regions around the longwave (LW) $CO_2$ sounding band (<800 cm$^{-1}$), $O_3$ band (980–1080 cm$^{-1}$), $H_2O$ band (1200–2150 cm$^{-1}$), and the shortwave (SW) $CO_2$ band (2200–2400 cm$^{-1}$). The sensitivity of these 'channels' with the largest additional uncertainties are shown for each band in Figure 10. This figure shows the broad nature of the temperature weighting functions in these regions. There is sensitivity to temperature above and below the 30 hPa radiosonde cut-off, indicating the importance of accurate temperature profiles at these high
altitudes. While the additional uncertainty is less for the ERA topped profiles in both $CO_2$ bands and the $O_3$ band, it is about the same in the $H_2O$ band. This indicates that the while ERA topped profiles mitigate most, but not all, of the impact from the truncated temperature information, it is not able to make up for the loss of the water vapor information. This shows the importance of both temperature and water vapor measurements at high altitudes for validation of sounding channels on IASI and other similar IR sounders.






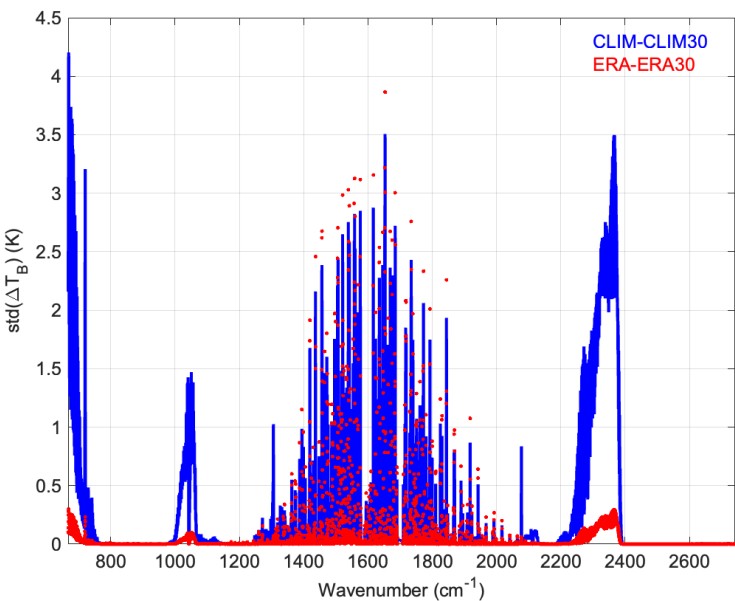

**Figure 9: Additional random uncertainty ($std(\Delta T_B)$) over the IASI wavenumber range and resolution for profiles topped with climatology (CLIM) in blue and with ERA5 (ERA) overlaid in red.**

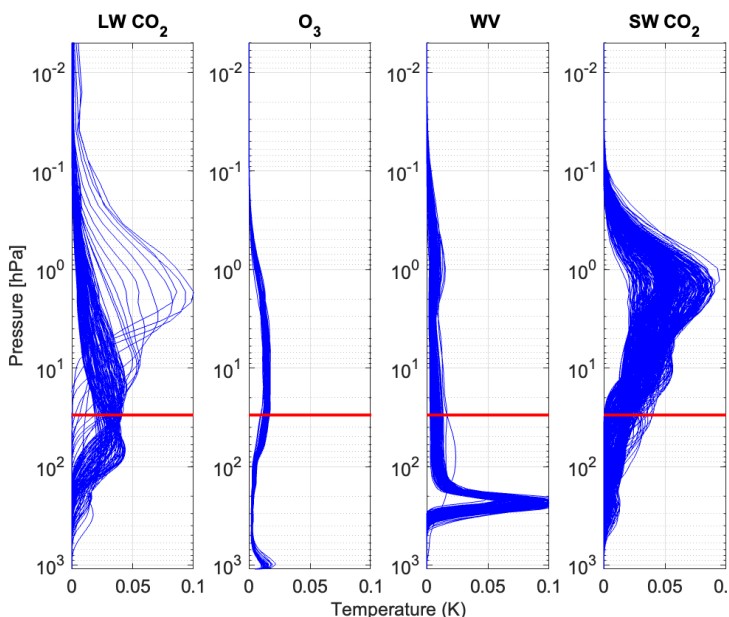

**Figure 10: IASI weighting functions in temperature sensitive wavenumber regions including the LW $CO_2$ (left), $O_3$ (2nd subplot), $H_2O$ (3rd subplot), and the SW $CO_2$ (right) regions. The red horizontal line is drawn at the 30 hPa pressure level.**





## 4.2 Comparison of various GNSS temperature retrieval data sets with GRUAN radiosonde data

GNSS RO measurements are assimilated as bending angle or refractivity, providing information on temperature, pressure, and humidity in recent operational analysis systems and reanalysis systems (e.g. Chapter 2, Section 2.4 of SPARC, 2022; Ruston et al., 2022). Hersbach et al. (2020, their Section 5.8) note that GNSS RO bending angles can be assimilated without bias correction that is needed for assimilation of satellite radiances (see also Section 5.1). The accuracy of their temperature data in the upper troposphere and lower stratosphere may be as good as those from the best radiosonde models (e.g. Tradowsky, 2019; Tradowsky et al., 2017). On the other hand, there are several institutes that provide GNSS RO temperature retrievals (see e.g. Steiner et al., 2020a), which show differences in particular above ~25 km due to differences in the processing algorithms, e.g. treatments of the ionospheric effects (Danzer et al., 2015). Reference-quality radiosonde data covering 30 hPa to e.g. 5 hPa are essential in the cross-validation of various GNSS RO temperature retrieval data products.

In this section, we investigate differences among five GNSS RO temperature retrieval data products (Table 1) and RS92 radiosonde GRUAN data product RS92-GDP.2 at Lindenberg, Germany (52.21°N, 14.12°E; LIN) and at Tateno, Japan (36.06°N, 140.13°E; TAT). The original GNSS RO data are those from Formosa Satellite 3/Constellation Observing System for Meteorology, Ionosphere and Climate (COSMIC Mission #1; Anthes et al., 2008). With an orbit inclination of 72°, the RO events of the six COSMIC-1 satellites are mostly concentrated around the midlatitudes of both hemispheres (see Son et al., 2011). By considering the high density of RO events at midlatitudes, we collected 14,217 and 1,946 profiles of the RS92-GDP at LIN and TAT sites, respectively, between 2007 and 2019. The collocation conditions are set as within 3° longitude/latitude and ±3 hours. The number of collocation pairs is, on average among the five GNSS RO data products, 3,000 and 200 at LIN and TAT, respectively. The vertical coordinate for the comparison here is geopotential height. Each individual temperature profile, both from radiosondes and from RO, has been averaged for 1 km at every 1 km step from 10 km to 40 km height range (for example, data from 31.5 km to 32.5 km are averaged to create 32 km data). We chose 1 km averaging here because the effective vertical resolution of COSMIC temperature retrieval data products may vary between 0.2 km and 1 km around 30 km (Tsuda et al., 2011). It is noted that most retrieval institutes provide two different temperatures, i.e. dry temperature (Tdry) and wet temperature (Twet). In general, in the upper troposphere and in the stratosphere, Tdry data may be more accurate than Twet because the former are directly derived from the RO refractivity while derivation of the latter involves one-dimensional variational (1D-Var) analysis method and thus a model (Sun et al., 2013). Therefore, in the current study, we use Tdry data where available, i.e. for the UCAR, WEGC, and RISH datasets, while we use Twet data for the ROMSAF dataset, which only provided Twet when we obtained it. For the JPL dataset, it is unclear from the data file header whether it is Tdry or Twet, but the lower tropospheric profiles suggest that it is Twet. Comparison of UCAR Tdry and Twet at the LIN site shows that Twet



is ~0.02 K smaller than Tdry between ~15 km to 35 km (not shown); therefore, we believe that the choice of Tdry vs. Twet is irrelevant in this altitude region.

Figures 11 and 12 show the mean $\Delta T$ (= $T_{RO}$ – $T_{GDP}$) profiles at LIN and TAT sites, respectively. Up to ~30 km, the differences are small (within ±0.3 K at both sites) for all the five GNSS RO data products, being consistent with the results demonstrated
by Steiner et al. (2020a). Above 30 km, the differences tend to be greater at higher altitudes. The difference using the UCAR data tends to be positive, and that of the WEGC and the RISH data tend to be negative at both sites. The tendencies among different GNSS RO temperature retrieval products obtained here are not necessarily consistent with Steiner et al. (2020a) (in their Figs. 2 and 3). This is probably due to the fact that Steiner et al. (2020a) compared the same RO events with different retrieval algorithms, while we compared GNSS RO events with radiosonde flights within a certain distance (i.e. up to 3° in
both latitude and longitude). We tested <2° for the collocation condition, and found that the tendencies to be essentially unchanged. However, as shown in the profiles of standard deviation and number of collocation pairs, the number of pairs decreases significantly above 30 km because of the limited height attainment by the radiosondes. Note that a peak in standard deviation at 14–19 km at TAT site shown in Fig. 12 is due to a large seasonal cycle of tropopause height (Noersomadi and Tsuda, 2017). The results shown here as well as those by Steiner et al. (2020a) clearly indicate the need for reference-quality
radiosonde temperature data above 30 km to validate various different GNSS RO temperature retrieval data products.

**Table 1: The COSMIC GNSS RO temperature retrieval data products analysed in this study.**

| Institute;<br>Data URL | Data version;<br>Tdry or Twet | Period<br>(Year.Day of year) | Reference |
|---|---|---|---|
| UCAR: University Corporation for Atmospheric Research, USA;<br>https://data.cosmic.ucar.edu/gnss-ro/cosmic1/repro2021/<br>(last access: 25 Jan. 2024) | wetPf2 version 2021;<br>Tdry | 2007.001 – 2019.344 | Wee et al. (2022) |
| JPL: Jet Propulsion Laboratory, NASA, USA<br>https://genesis.jpl.nasa.gov/data/ftp/<br>(last access: 25 Jan. 2024) | version 2.6;<br>Twet | 2007.001 – 2019.344 | Hajj et al. (2002) |
| WEGC: Wegener Center, Univ. of Graz, Austria<br>https://wegcwww.uni-graz.at/data-store/WEGC/OPS5.6:2021.1/<br>(last access: 25 Jan. 2024) | OPS version 5.6;<br>Tdry | 2007.001 – 2018.365 | Angerer et al. (2017) |
| RISH: Research Institute for Sustainable | IUGONET Version 1.0; | 2007.001 – 2017.089 | Tsuda et al. (2011) |





| Humanosphere, Kyoto University, Japan http://database.rish.kyoto-u.ac.jp/arch/iugonet/GPS/index.html (last access: 25 Jan. 2024) | Tdry | | |
|---|---|---|---|
| ROMSAF: Radio Occultation Meteorology Satellite Application Facility, European Organisation for the Exploitation of Meteorological Satellites https://rom-saf.eumetsat.int/product_archive.php (last access: 25 Jan. 2024) | Version 1.0; Twet | 2007.001 – 2016.366 | https://rom-saf.eumetsat.int/product_documents.php (last access: 25 Jan. 2024) |


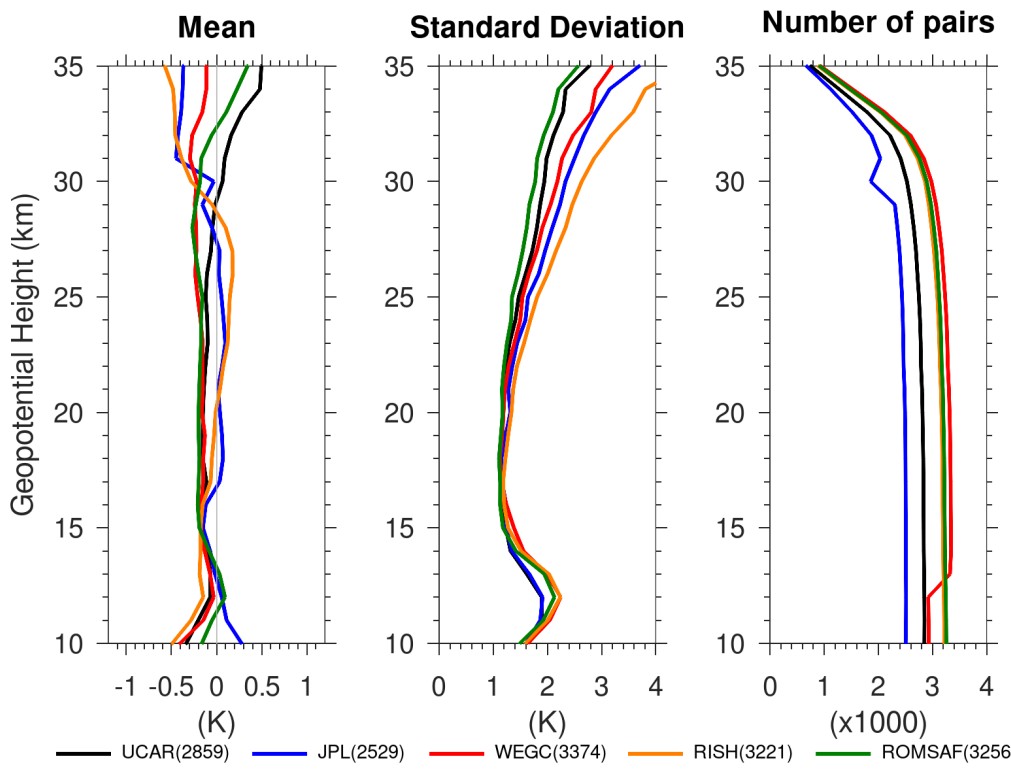

**Figure 11: $\Delta T$, standard deviation, and the number of collocation pairs between five GNSS RO temperature retrieval products for the COSMIC and radiosonde GRUAN data product RS92-GDP.2 at Lindenberg (LIN), Germany.**



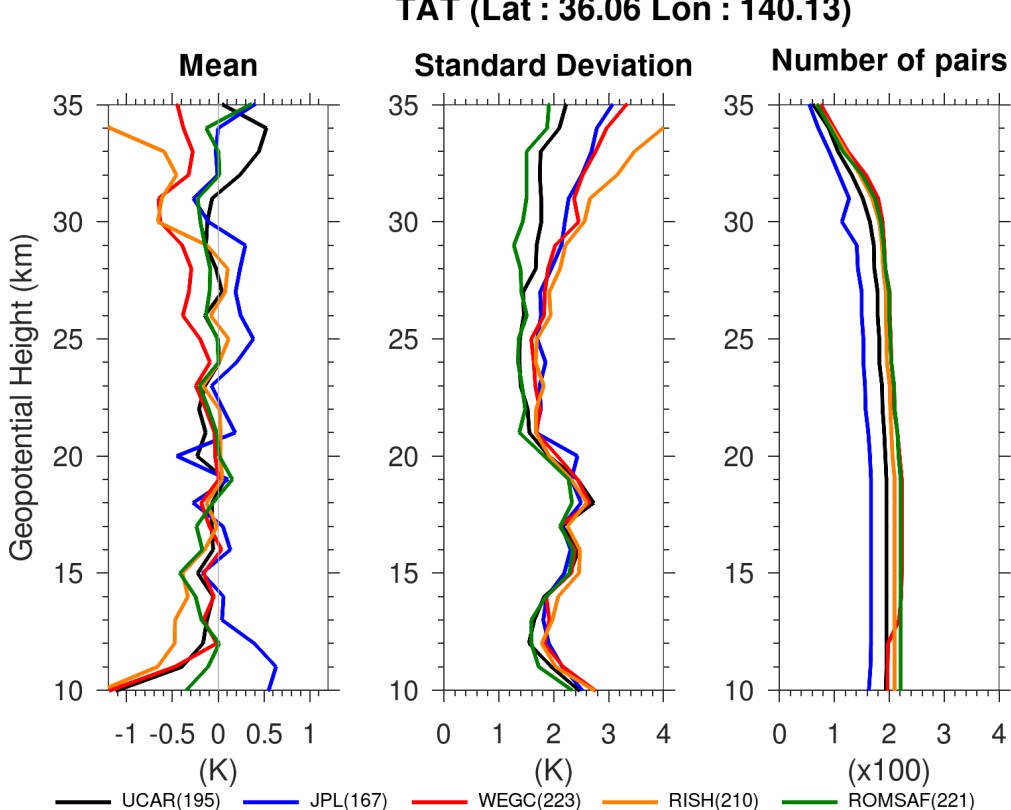

Figure 12: As for Fig. 11 but for Tateno (TAT), Japan.


## 5 Impacts on numerical weather forecasts

### 5.1 Role of "anchor" observations in numerical weather prediction systems

NWP systems use data assimilation methods to produce the best estimate of the atmospheric state, the analysis, from which forecasts are initiated (Rodgers, 2000). Obtaining the analysis is a Bayesian problem, i.e. finding the probability of the atmospheric state, $x$, given the observations, $y$. In practice, the peak of the probability density function (PDF) is calculated, i.e. the most probable value of $x$ given $y$, solved through the minimization of the cost function $J(x)$ which is defined as:

$$2J(x) = (x - x^b)^T B^{-1} (x - x^b) + (y - H(x))^T R^{-1} (y - H(x)) \tag{1}$$

where $y$ is the observation such as $y = H(x^t + \varepsilon^{obs})$, $x^b$ is the background (a short-range forecast) such as $x^b = x^t + \varepsilon^b$ with $x^t$ being the true value, $H$ is the observation operator, $\varepsilon^{obs}$ and $\varepsilon^b$ are the errors in the observation and the background,





respectively, and $R$ and $B$ are the observation and background error covariances, respectively. The left-hand term of Eq. (1) is referred to as the background penalty, often noted $J_b$, while the right-hand term is referred to as the observation penalty, or $J_o$. A solution is obtained when the gradient of the cost function reaches its minimum through an iterative process.

Data assimilation theory, however, assumes a Gaussian PDF, free from systematic error. Therefore, bias correction schemes are employed to correct systematic errors in satellite observations during the data assimilation cycle by adding a correction term to the observation operator (e.g. Dee and Uppala, 2009; Laloyaux et al., 2020; Hersbach et al., 2020).

Problems arise when the model biases are not negligible because it can be difficult to disentangle model biases from biases in the observations. Left unchecked, the bias correction gradually corrects observations towards model biases and propagates biases in the analysis. Anchoring the bias correction is possible when unbiased observations (or with a bias much smaller than that of the model) are assimilated uncorrected within the variational scheme (although they may be corrected in e.g. an independent pre-processing step). These "anchor" observations can only mitigate the contamination of the bias correction by the model biases but not remove it completely (Eyre, 2016; Francis et al., 2023). The situations where anchors are useful and situations where they are not can be summarised as follows:

- If model biases are present in state variables observed by anchor observations but not observed by bias-corrected observations, and these model state variables are strongly correlated (for example, where smaller-scale errors dominate, as in clouds and precipitation fields, the error can be correlated with errors in other state variables over shorter distances), then the anchor observations cannot reduce the contamination of the bias correction.
- If model biases are present in state variables observed by bias-corrected observations but not observed by anchor observations, then the anchor observations cannot reduce the contamination of the bias correction.
- If model biases are present in state variables observed by both anchor and bias-corrected observations, then the anchor observations can mitigate the contamination of the bias correction.

In order for the latter to be true, the error covariance ($R$ in Eq. (1)) assigned to anchor observations in the data assimilation system must be smaller than that of the background. The greater the weight of the anchor observation (i.e. the smaller the covariance error), the more it will contribute to mitigating the contamination. The observation covariance error ($R$ in Eq. (1)) and therefore the weight given to the observation is tied to observation uncertainty and observation density. In other words, in order to maximise the reduction of model bias contamination, high-quality and high-spatial and temporal coverage anchor observations are needed. Most modern NWP systems utilise radiosondes, GNSS RO, and some satellite radiance observations to anchor the bias correction in relation to different state variables.

Radiosonde profiles offer accurate vertical sampling of the troposphere and the lower stratosphere, although the coverage is largely limited to land masses, predominantly in the northern hemisphere, except for a few radiosondes launched from ships



and islands. The solar radiation-induced bias is considered the primary source of error for radiosonde temperature (Section 2), but it is typically corrected for prior to being incorporated into the NWP systems (von Rohden et al., 2022). Radiosonde
measurements therefore present the required characteristics to anchor the bias correction, albeit not at global scale, of tropospheric temperature, humidity, and wind, and of stratospheric temperature and wind. (The radiosonde humidity is typically systematically underestimated (Section 2) and quality controlled out of NWP systems at high altitudes.)

In contrast, GNSS RO observations have the benefit of providing a good global coverage of the upper tropospheric and
stratospheric temperatures. Furthermore, they do not require prior correction for the NWP systems as the bias in the bending angle is much smaller than model biases. Satellite radiances from high-peaking temperature sounding channels, such as ATMS channel 15 (57.290344 ± 0.3222 ± 0.0045 GHz, peaking around 2 hPa), also provide global coverage and present biases that are sufficiently small compared to model biases that they can be assimilated without correction. These observations can therefore act as anchor with respect to temperature but cannot mitigate the contamination of the bias correction by other state
variables.

## 5.2 Global radiosonde data denial experiments using a Met Office NWP system

In preparation for the 8th WMO Workshop on the Impact of Various Observing Systems on NWP and Earth System Prediction
(ESP) (https://community.wmo.int/en/meetings/8th-wmo-impact-workshop-home, last access: 17 May 2024), a series of global data denial experiments have been conducted at the Met Office. The workshop is conducted every four years and aims at providing scientific evidence on the impact that the various components of the Earth observing system are having on forecasts and climate monitoring. The results of this workshop inform the preparation of WMO policies and their recommendations for future observing systems and provide valuable guidance to meteorological services on the optimal use
of the current observing system. In preparation for the workshop, WMO identifies a number of scientific questions to be addressed as a matter of priority. The work presented in this section aligns with the question related to Surface-Based Observing Systems (S1) Sub-Type Radiosondes (S1.2): *What is the impact of high-altitude ascent and descent data of radiosondes on forecast skills (weather, climate, etc.)?*

The experiments conducted at the Met Office have been configured as a low-resolution version of the operational global NWP system decoupled from ocean and surface (i.e. atmosphere only). Forecasts are produced by the Unified Model at N320L70 resolution (~40 km grid length and 70 levels with the model top at 80km) and the data assimilation uses a hybrid incremental four-dimensional variational (4D-Var) analysis method of dual resolution N108/N216L70 (~120/~60 km, 70 levels). The system uses a flow-dependent background error associated to a N216L70 44-member 9-hour forecast ensemble (Lorenc et al.,





2000; 2015; Rawlins et al., 2007). The observing system used for data assimilation reflects operational practice during winter 2022–2023. The control and subsequent experiments cover the period from December 15, 2022 to March 15, 2023. The evaluation of all experiments against the control run is done from December 22, 2022, allowing the bias correction to spin up for seven days. Note that most of the radiosonde observations assimilated at Met Office are ascending profiles, although descents are also assimilated over Germany. A total of four experiments are discussed below. They are referred to as NS (No

Sonde), NRO (No GNSS RO), NS30 (No Sonde beyond 30 hPa), and NS30CV (NS30 with Control VarBC).

In a first experiment, we remove all radiosondes from assimilation, hereafter referred to as NS (No Sonde). This includes profiles of temperature, relative humidity, and wind. The root-mean-square error (RMSE) calculated against a proxy of truth is compared to that of the control run for a selection of forecast variables (wind, temperature, and geopotential height) in the

northern hemisphere, tropics, and southern hemisphere at different pressures (from the surface to 50 hPa) and lead times (from analysis time plus 12h, T+12, to T+144). Proxies of truth are independent data sources, typically ECMWF analysis and observations (e.g. sondes, aircraft, surface, and satellite-based wind). A performance score is then obtained from the overall change in RMSE over the length of the experiment. For an impact to be considered significant in the three-month-long NWP experiments at the Met Office, it was estimated that a difference of at least ±0.1 % RMSE compared to the control is required.

This is a standard evaluation metric used operationally at the Met Office to evaluate the performance of observing system experiments.

From the NS experiments, we found that the overall RMSE increases by 1.66 % and 1.34 % against ECMWF and observations, respectively, due to the exclusion of the radiosonde data. RMSE changes are reported in Table 2. This represents a major

degradation in performance. Compared to other denial experiments (not detailed here), radiosondes were found to be the third most important component of the observing system after GNSS RO and the space-borne microwave instruments when using observations as proxy of truth, and the fourth (after space-born infrared instruments) when using ECMWF analysis as proxy of truth. A detailed analysis of individual change of RMSE reveals that all analysed variables degrade to some extent with the largest changes observed in the tropospheric (in the 850–250 hPa range) northern hemisphere at short lead times. The larger

degradation detected in the northern hemisphere is likely related to the choice of the season over which the experiments are conducted, boreal winter, which finds most of impactful weather driving higher scores. The larger degradation at short lead times (typically largest at T+12) evidences the lessening impact of the data assimilation system as the forecasts range increases.

Is it then useful for NWP systems to have radiosondes that reach a pressure of 30 hPa or less? In a second experiment, we try

to address this question with radiosonde profiles cut at 30 hPa; they remain assimilated at pressures greater than this threshold but excluded at and below it. This experiment will be referred to as NS30 (No Sonde beyond 30 hPa). In this case, the overall change in RMSE, reported in Table 2, does not reach significant levels, i.e. the impact is neutral. However, it must be stressed that this metric is not designed to evaluate changes in the mid and upper stratosphere as, indeed, the lowest pressure being





considered for the calculation of the RMSE is 50 hPa. To evaluate the impact of removing upper radiosonde profiles in the
stratosphere, we calculate the relative change in RMSE in the analysis (T+0) against independent radiosondes at 10 hPa.
Statistics for the northern polar region (from 90 to 60° N), northern hemisphere (from 90 to 18.75° N), Tropics (from 18.75°
N to 18.75° S), southern hemisphere (from 18.75 to 90° S), and southern polar region (from 60 to 90° S) are shown in Table
3. The RMSE is shown to increase in all latitude bands, from 1.76 % to 6.57 % for temperature and from 6.14 % to 11.81 %
for wind. The largest degradation is observed in both cases in the northern hemisphere, again likely in relation to the season
over which the experiment has been conducted. In subsequent forecasts, the RMSE difference between control and experiment
tends to converge towards a value close to zero. This is illustrated in Figure 13, where both RMSE and mean error in the
experiment and the control (and their respective difference) are shown for the wind at 10 hPa in the northern hemisphere from
T+0 to T+144. We note that the mean error remains relatively constant beyond T+24 which suggests the presence of a model
bias as the data assimilation will have little impact over these long lead times.


For comparison, results from an experiment where all GNSS RO data are removed from assimilation, noted as NRO, are shown
in Table 3. The change in RMSE for temperature is of the same order as for NS30, but the largest change is found in the tropics
(4.97 %). For wind, the change is, without surprise, much smaller, since GNSS RO do not provide wind measurements. The
RMSE difference with respect to the control is one order of magnitude smaller than in NS30, except in the tropics where it
reaches 6.86 %. The larger impact of GNSS RO in the tropics is likely related to the greater number of available and assimilated
GNSS RO observations compared to mid and high latitudes. The extra number of observations is driven by the COSMIC
Mission #2 (COSMIC-2) constellations that observe Earth from low-inclination orbits (Schreiner et al., 2020).




**Table 2: Overall relative RMSE change against ECMWF analysis and observations obtained from 576 forecast variables (wind,**
**temperature, and geopotential height for different latitude bands, pressures, and lead times) between December 22, 2022 to March**
**15, 2023.**

| Experiments | RMSE against ECMWF analysis (%) | RMSE against observations (%) |
|---|---|---|
| NS | 1.66 | 1.34 |
| NRO | 2.78 | 1.44 |
| NS30 | 0.02 | 0.04 |
| NS30CV | 0.02 | 0.0 |

**Table 3: Relative change in RMSE for temperature and wind (U-component) at 10 hPa in the analysis (T+0) against independent**
**radiosondes. Statistics are shown for five latitude bands. For each variable, NS30 and NS30V are compared to the control (Ctrl) and**
**against each other. NRO is compared to the control only.**

| | Temperature | | | | Zonal Wind | | | |
|---|---|---|---|---|---|---|---|---|
| | NS30 vs Ctrl (%) | NS30CV vs Ctrl (%) | NS30 vs NS30CV (%) | NRO vs Ctrl (%) | NS30 vs Ctrl (%) | NS30CV vs Ctrl (%) | NS30 vs NS30CV (%) | NRO vs Ctrl (%) |
| Northern Polar Region (90N-60N) | 5.76 | 5.53 | 0.21 | 4.21 | 8.93 | 8.95 | -0.02 | 0.24 |
| Northern Hemisphere (90N-18.75N) | 6.57 | 6.60 | -0.03 | 1.79 | 11.81 | 11.75 | 0.05 | 1.11 |
| Tropics (18.75N-18.75S) | 3.11 | 2.92 | 0.18 | 4.97 | 7.31 | 7.21 | 0.10 | 6.86 |
| Southern Hemisphere (18.75S-90S) | 1.76 | 2.05 | -0.29 | 0.25 | 6.14 | 6.35 | -0.20 | 0.61 |
| Southern Polar Region (60S-90S) | 2.20 | 2.62 | -0.41 | 4.36 | 7.93 | 8.15 | -0.20 | 0.89 |





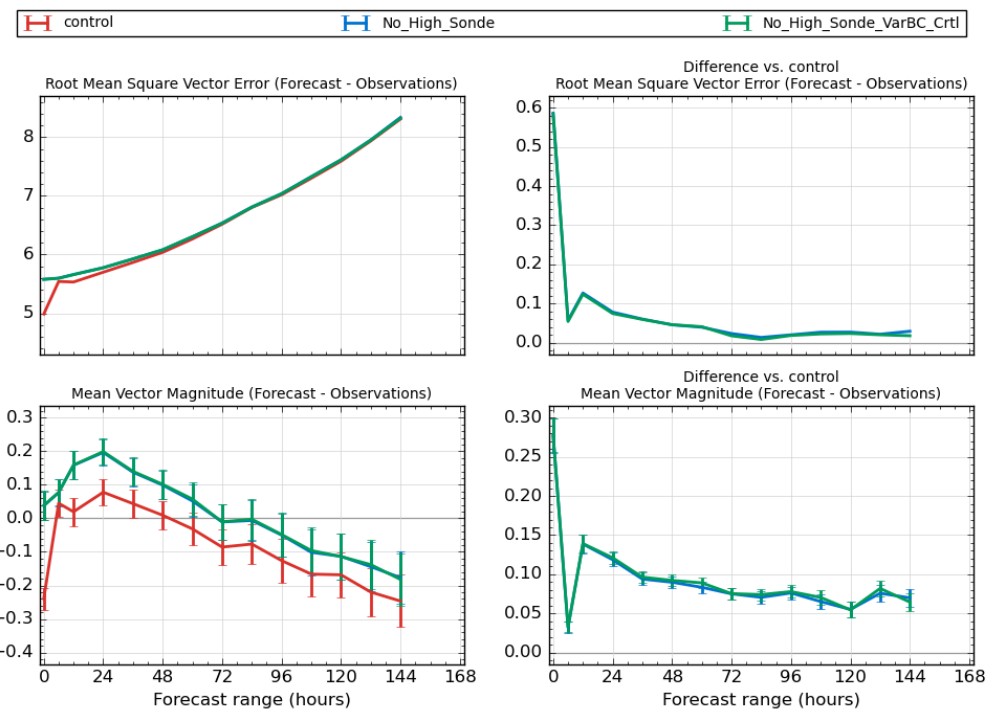

**Figure 13: RMSE against independent radiosondes in the control, NS30, and NS30CV experiments in the wind at 10 hPa in the northern hemisphere (top left) and absolute difference experiments and control (top right). Absolute mean error against independent radiosondes (bottom left) and difference against the control (bottom right).**

A second standard metric used to evaluate experiments is the fit between independent observations and the model background (i.e. the 6-h forecast used as first guess in 4D-Var), expressed as the variation in the standard deviation of the background departure. This metric has the advantage of providing insight into the atmospheric layers of interest on a channel-by-channel basis for the satellite instruments used in the data assimilation system. For the NS30 experiment, small but significant changes (i.e. all the reported values are withing the 95% confidence interval), of the order of a few tenths of a percent are detected for channels whose Jacobians peak between 90 hPa and 5 hPa. For example, up to 0.3 % degradation is detected for ATMS channels 10–14 (see Figure 6 for channel peaking pressure). Some instruments, however, present a reduction in the standard deviation of the same order in atmospheric layers similar to those covered by the ATMS channels 10–14. The Special Sensor Microwave Imager/Sounder (SSMIS) (Kunkee et al., 2008) sees the fit of channels 6 and 7, respectively peaking at 60 hPa and 35 hPa, degrading by 0.40 % but the fit of channels 23 and 24, respectively peaking at 5 hPa and 15 hPa, improving by 0.30 %.



Similar mixed results are also observed for other instruments across the microwave and infrared domains. It is not clear in such cases whether the reduction in standard deviation results from compensating biases or an actual improvement of the background after the removal of the radiosonde profiles. In either case, the changes are sufficiently small to be considered as neutral.

Finally, we investigate how the bias correction (rather than the error in the analysis and forecasts) is responding to the removal of radiosonde profiles in the stratosphere. As explained previously, the contamination of the bias correction by model bias can be mitigated where both the anchor observations, here the radiosondes, and bias-corrected observations, e.g., the satellite radiances, observe the same biased state variables. At the Met Office most assimilated satellite radiances are bias-corrected by a variational bias correction (VarBC) scheme similar to that described by Auligné et al. (2007). A detailed implementation of VarBC at the Met Office is provided by Cameron and Bell (2018).

Following the nomenclature used by Francis et al. (2023), with $x$ the NWP model state, $p$ the bias predictors, and $\beta$ the bias-correction coefficients, the bias correction term $c$ for the $k^{th}$ observation can be written as:

$$c_k = s_k + \sum_{i=1}^{r_k} \beta_{k,i} p_{k,i}(x) \tag{2}$$

where $s_k$ is a constant term and $r$ is the number of predictors.

Up to 31 predictors can be used in the bias correction scheme in place at the Met Office. The first predictor is a constant offset, i.e. $p_{k,0} = 1$. Two predictors are related to the thickness of the air-mass in the intervals 850–300hPa and 200–50hPa. One predictor uses the skin temperature. Another predictor uses the total water column. Bias variations across the scan of a satellite instrument can be represented by up to six Legendre polynomials. And bias along the orbit can be represented by up to 20 predictors forming the cosine and sine of a Fourier series. The coefficients $c_k$ are updated at each assimilation cycle during the variational analysis by an increment that minimises the difference from the analysis.

The evolution of the bias-correction coefficients in NS30 is compared to that of the control. Interestingly, we find that the coefficient of most satellite channels peaking in the stratosphere display various degrees of drift. Figure 14 illustrates the total change (left) in the coefficients across ATMS seven predictors (one predictor as a constant offset, two predictors of air-mass thickness in the intervals 850–300hPa and 200–50hPa, and four predictors of Legendre polynomials representing the variation in bias across the scan) and the specific change in the coefficient of the constant predictor for channel 12 (right). Channel 12, which peaks at 25 hPa, displays the largest difference with the control. The total change is predominantly driven by the drift of the constant predictor coefficient since none of the others would respond to a change in the stratosphere. At the end of the 3-month experiment , the coefficient for the constant predictor has drifted by 2 % compared to the control. This is, for comparison, similar to the drift observed in the denial experiment removing GNSS RO observations (not shown). The GNSS




RO experiment however presents a greater drift of the constant predictor coefficient for channels 11 and 13 (4.3 % and 4.6 %, respectively, compared to 0.6 % and 1.2 % for the NS30 experiment) that peak at 50 hPa and 10 hPa, respectively. The change in the coefficients is also generally larger in the GNSS RO experiment for other instruments compare to NS30.

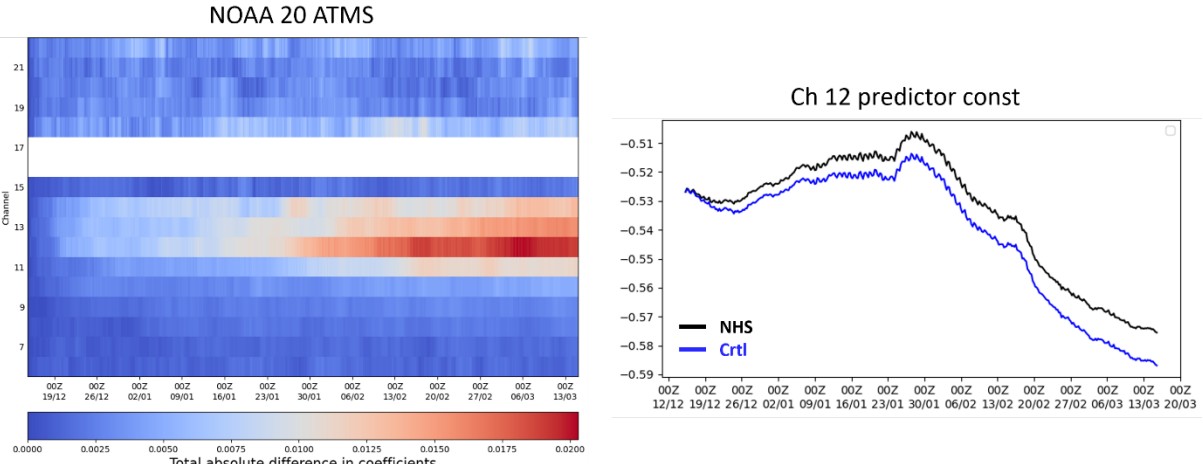

**Figure 14: (Left) total absolute difference compared to the control across all predictors used for NOAA 20 ATMS assimilated channels over the course of NS30. (Right) ATMS channel 12 constant predictor from the control (blue) and NS30 (black).**

Next, we investigate if the drift observed in the coefficients could be related to a drift towards model bias as predicted by Eyre (2016) and Francis et al. (2023) due to the loss of anchor observations or be related to a feedback loop caused by the degradation of the analysis. To address this point, we have conducted a third experiment, similar to NS30, but with the bias correction coefficients taken from the control (i.e. not updated against the experiment's analysis and oblivious of the missing radiosonde profiles), hereafter referred to as NS30CV (NS30 with Control VarBC). With this configuration, one would expect the suboptimal bias correction (which coefficients come from the control and not the experiment lacking radiosondes) to wrongly correct satellite channels peaking in the stratosphere. Here we do not mean the correction is wrong with respect to the (unknown) truth but rather in the sense that it does not correct against NS30CV's analysis. The background fit to observations would then degrade and ultimately impact the new analysis. If none of these is observed, a drift resulting from the lack of anchor observations becomes the most likely hypothesis.

RMSE changes for temperature and wind at 10 hPa against the control and against NS30 are reported in Table 3. The results are very similar to those of NS30 with differences no larger than 0.4 %. The background fit to observations is also comparable to NS30 (not shown). The similar results obtained from NS30 and NS30CV experiments suggest that the analysis does not change significantly with or without suboptimal bias correction. We can therefore conclude that the observed drift in the bias



coefficients is not related to VarBC adapting to the new analysis. As hypothesised above, the reduction in number of anchor observations resulting from the removal of the upper part of the radiosonde profiles provides the best explanation for the bias coefficient drift. It is therefore important to stress that over time, such drift would inexorably degrade the quality of the bias correction, and by extension the quality of the data being assimilated. Although no significant impact was detected on a 3-month timescale, it is reasonable to expect long-term analyses and forecasts degradation resulting from the removal of anchor

observations.

In summary, we have demonstrated through a dedicated set of denial experiments that radiosonde profiles play a crucial role in the observing system utilised by the Met Office NWP global model. Although most of the impact is observed in the troposphere, we were also able to demonstrate that removing radiosonde profiles at pressures less than 30 hPa causes the 10

hPa wind and temperature RMSE in the analysis to increase significantly, with the largest degradation observed in the winter hemisphere. The wind analysis is impacted nearly twice as much as temperature, highlighting the importance of radiosondes to constrain wind in the mid-stratosphere. The background fit to satellite channels peaking in the stratosphere was found to be broadly neutral, but it was shown that the bias-correction coefficients have started drifting away from those in the control when radiosondes were not used to anchor the bias correction. Coefficient drift did not cause visible detrimental effects over the

length of the 3-month experiments. However, in the long term, it could become a major concern as it may undermine the value of these satellite instruments, degrade the quality of the assimilated data, and ultimately impact both analysis and forecasts with potential repercussions propagating to lower levels in the troposphere and lower stratosphere. Finally, experiments with removed GNSS RO observations from assimilation have helped identify the relative contribution of radiosonde profiles for anchoring the satellite bias correction in the stratosphere: The contribution of radiosonde observations at pressures less than

30 hPa is significant but of lesser amplitude than observations from GNSS RO. Note that these conclusions should be considered with caution due to the limited length of the experiments. Longer experiments (e.g., 9 or 12 months) would provide more comprehensive and statistically robust results. However, this was not achievable given resource and time constraints.

**5.3 Value of high-ascent radiosondes in NWP data assimilation and forecast system**

NWP impact studies, such as the ones conducted in the previous section 5.2 and by Bormann et al. (2019) are the methods normally used to assess if the individual observing systems, including satellite radiance measurements, GNSS RO measurements, and conventional observations, contribute to reducing the short-term forecast error. The previous section 5.2 revealed that conventional radiosondes play an important role in the NWP short-term forecasting. The study further indicated

that radiosonde profiles reaching 30 hPa or higher tend to have a significant impact on the wind analysis in the stratosphere. The impact of high ceiling radiosondes reaching 10 hPa or higher, however, are not specifically addressed in this and other



similar studies (e.g. Bormann et al., 2019). The reason for that is perhaps the number of high-ascent radiosondes (i.e. about 300 profiles daily) is limited and it is challenging to detect their impact, if any, on a global scale.

In this section, we analyse the NWP analysis fields, such as air temperatures, associated with high-ascent and low-ascent radiosondes, respectively, to investigate if they differ from each other. And from there, a question will be asked if high-ascent radiosondes contribute to the analysis field, thus influencing the NWP data assimilation and forecasting, a topic deserving further investigation. Temperature analysis data generated from the Integrated Forecast System (IFS) model (ECMWF, 2018) of the European Centre for Medium-Range Weather Forecasts (ECMWF) are analysed for this purpose. Note that the NWP

analysis field combines short-term forecasts with observations to provide an atmospheric state, which is then used as the initial state in the NWP. We conduct this empirical data analysis by comparing the ECMWF analysis collocated with high ceiling radiosondes with the ECMWF analysis collocated with only low ceiling radiosondes. GNSS RO "dry temperature" (Tdry) data (Sun et al., 2019) are used as the transfer medium to assess the ECMWF analysis associated with high versus low-ascent radiosondes. In recent years various changes have been made to reduce stratospheric biases in the ECMWF analyses (Laloyaux

et al., 2020; and Polichtchouk et al., 2021).

Collocations of RAOBs with GNSS RO Tdry and ECMWF data for the period July 1, 2021 to February 26, 2023 collected via NOAA Products Validation System (NPROVS, Reale et al., 2012) are used in the analysis. Since radiosonde data accuracy may vary with radiosonde type (Sun et al., 2019), to keep the analysis robust, we select only collocations with Vaisala RS41

radiosondes (Dirksen et al., 2020), one of the most advanced radiosonde types available. RS41 sondes within 1 hr from ECMWF analysis are used. Note that it generally takes about 30 minutes for balloons to reach the upper troposphere (~300 hPa, see Seidel et al., 2011). To avoid the analysis bias towards the lower altitude, 30 minutes was therefore added to the radiosonde launch times when finding the collocations of radiosonde with ECMWF (or satellite retrieval) profiles (Sun et al., 2023). Since ECMWF analysis fields are at four synoptic times (with the main analysis being at 00 and 12 UTC), about 85 %

of all RAOB–ECMWF collocations are accepted for analysis. To reduce the bias and uncertainty that tends to increase with spatial and temporal collocation mismatch (Sun et al., 2019), we use only those collocations for which the ECMWF and RO are separated within 2 h (rather than 1 h for this case) and 150 km. Two sets of RAOB-ECMWF-RO collocations are analysed, those with RO Tdry from UCAR COSMIC-2 and from EUMETSAT ROM SAF MetOp GRAS products, respectively (see Sun et al., 2019 for details on the RO products).


In this study, we compare two subsets of RAOB-ECMWF-RO collocations: A) the RS41 balloons ascending to at least 8 hPa, and, B) the RS41 balloons ascending up to 30 hPa. The selection of 8 and 30 hPa is somewhat arbitrary, but the former case presumably represents high ascent balloons and the latter case represents low ascent balloons. Note that the collocations for Cases A and B are mutually exclusive. The numbers of collocation sample for 8 hPa and 30 hPa ascent RAOBs are comparable

to each other (see Figures 15 and 17, respectively), and this similarity also holds for day/night, which is also one of the



considerations for selecting those pressure threshold values for the analysis. The RAOB-ECMWF-RO COSMIC-2 collocations are restricted to within 45°N and 45°S where COSMIC-2 profiles are limited to (Figure 15). Figure 16 shows that ECMWF demonstrates a slight cold bias (~0.1 K) in the pressure height from 100 to 8 hPa for both Cases A and B relative to UCAR COSMIC-2 Tdry. ECMWF biases for these two cases match closely to each other from 100 up to 30 hPa. The ECMWF biases

for these two cases, however, depart from each other starting at 30 hPa and the departure increases with height, reaching ~0.1 K at 10 hPa, indicating that the temperature analysis associated with high-ceiling radiosonde observations tend to be warmer than the one associated with low-ceiling radiosondes in the low and mid-stratosphere.

Similar results are obtained from analysing the RAOB-ECMWF-RO GRAS collocations (Figures 17 and 18). By using GRAS

RO Tdry as the transfer medium, the temperature analysis associated with high-ascent radiosonde observations tends to be warmer than the one associated with low-ascent radiosondes by ~ 0.3 K at 10 hPa averaged from global sites where RS41 radiosondes are collocated (Figure 17).

 Different from the COSMIC-2  data which is limited to tropical and subtropical regions (Figure 15), GRAS has global data

coverage (Figure 17). For the high-ascent radiosonde case (i.e. Case A), relative to GRAS Tdry, ECMWF analysis exhibits a slightly warm bias (red curve in Figure 18), which is similar to the corresponding case (red curve in Figure 16) where COSMIC-2 is used as the target. As a matter of fact, Tdry from GRAS and COSMIC-2 is consistent with each other (< ~0.1 K) obtained by using the high-ascent radiosondes as the transfer medium. For the low-ascent radiosonde case (i.e. Case B) particularly for the pressure height above 30 hPa where radiosonde observations are not available, the ECMWF cold bias (blue curve) in Figure

17 is somehow inconsistent with what is shown in Figure 16 (blue curve), where ECMWF analysis is biased  slightly warm. Further investigation is needed to understand this inconsistency, but  it could  be related to their difference in data spatial domain  and consequently the difference in the observation data assimilated in the model.

Significant NWP forecast benefit has been obtained from assimilating data from global observing systems including

conventional radiosonde data (e.g., Bormann et al., 2019; Carminati et al., 2019). The impact of high-ascent radiosondes, however, is not specifically addressed in those studies. This empirical analysis indicates that the temperature analysis fields differ from each other for the cases with high-ceiling versus low-ceiling radiosondes. The temperature analysis associated with high-ceiling radiosondes tends to be warmer than the ones associated with low-ceiling radiosondes, their difference increasing (slightly) with height in the low-mid stratosphere, reaching 0.1–0.3 K at 10 hPa. Further study is needed to understand what is

the impact of that difference on short-term weather forecasting and long-term climate detection.



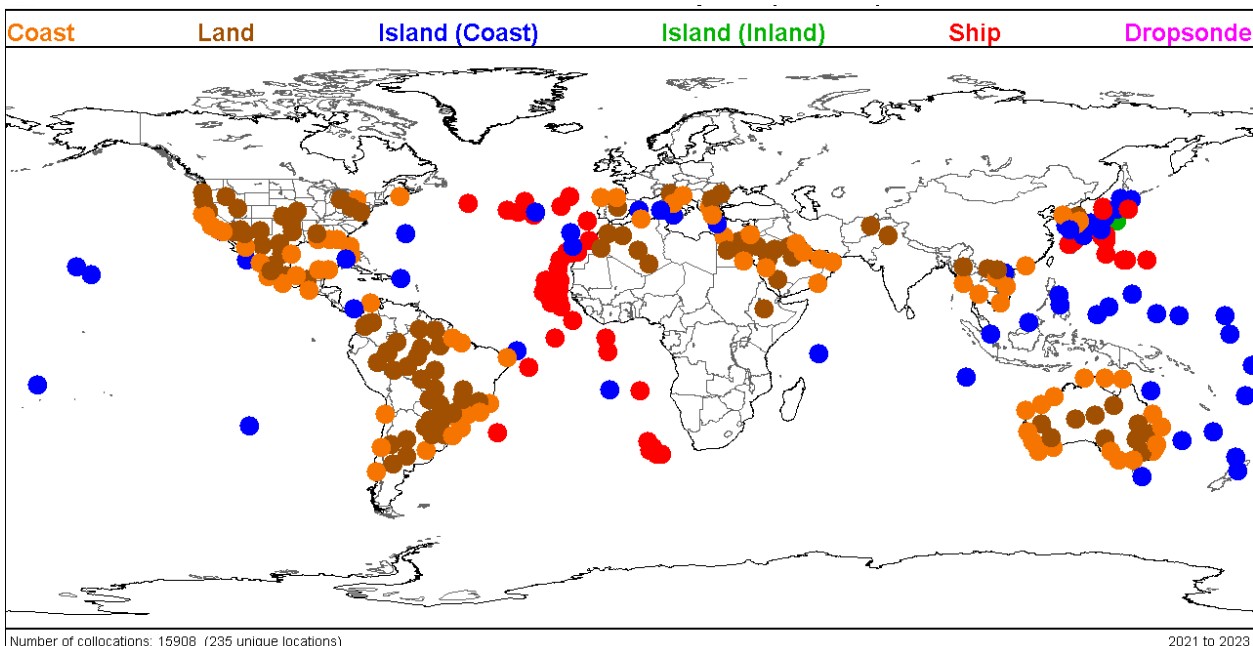

**Figure 15: Spatial distribution of Vaisala RS41 radiosonde data collocated with operational ECMWF analysis temperature and UCAR COSMIC-2 RO dry temperature (Tdry). See text for the collocation criteria. Data for the period 1 July 2021 to 26 February 2023 are used. Among the RS41 radiosonde profiles used (15,908), 22.9% reach at least 8 hPa, and 23.9% reach up to 30 hPa. Radiosondes are color-coded based on radiosonde terrain with brown (land), orange (coast), blue (island), green (island inland), and red (ship).**



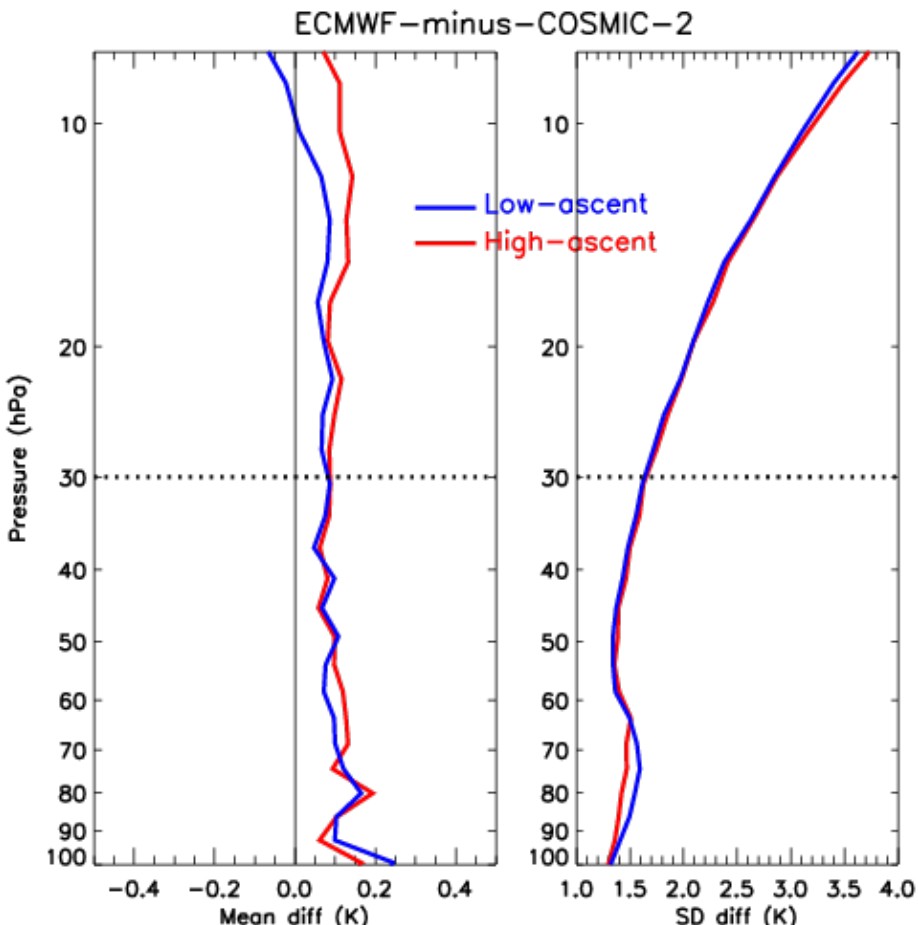

**Figure 16: (Left) Mean differences of ECMWF temperature analysis minus UCAR COSMIC-2 RO Tdry. Red curve denotes the difference associated with RS41 ascending to at least 8 hPa (labelled as "High-ascent"), and blue curve associated with RS41 radiosondes ascending up to 30 hPa (labelled as "Low-ascent"). (Middle) Standard deviations of the temperature differences. The number of collocated profiles for "High-ascent" and "Low-ascent" are ~3180 and ~3510, respectively.**



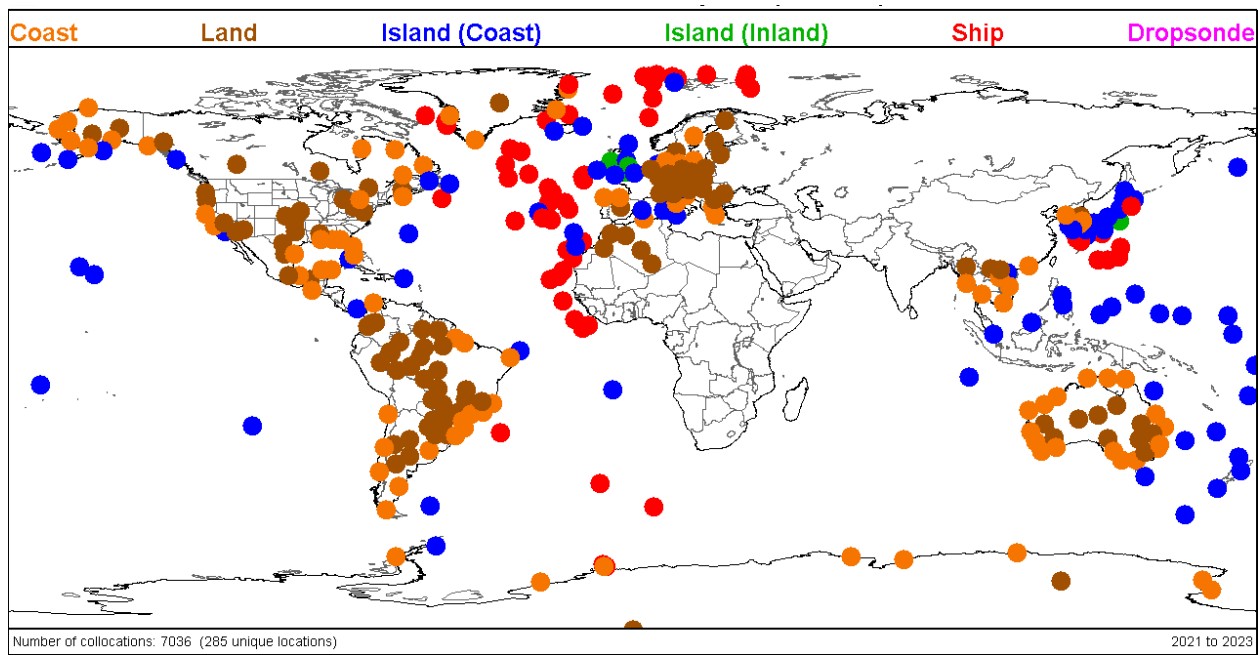

**Figure 17: As for Figure 15 but for collocations of RS41 radiosonde soundings with ECMWF and ROM SAF GRAS RO Tdry. Of 7,036 RS41 radiosonde profiles, 26.2% reach at least 8 hPa, and 20.8% reach up to 30 hPa.**





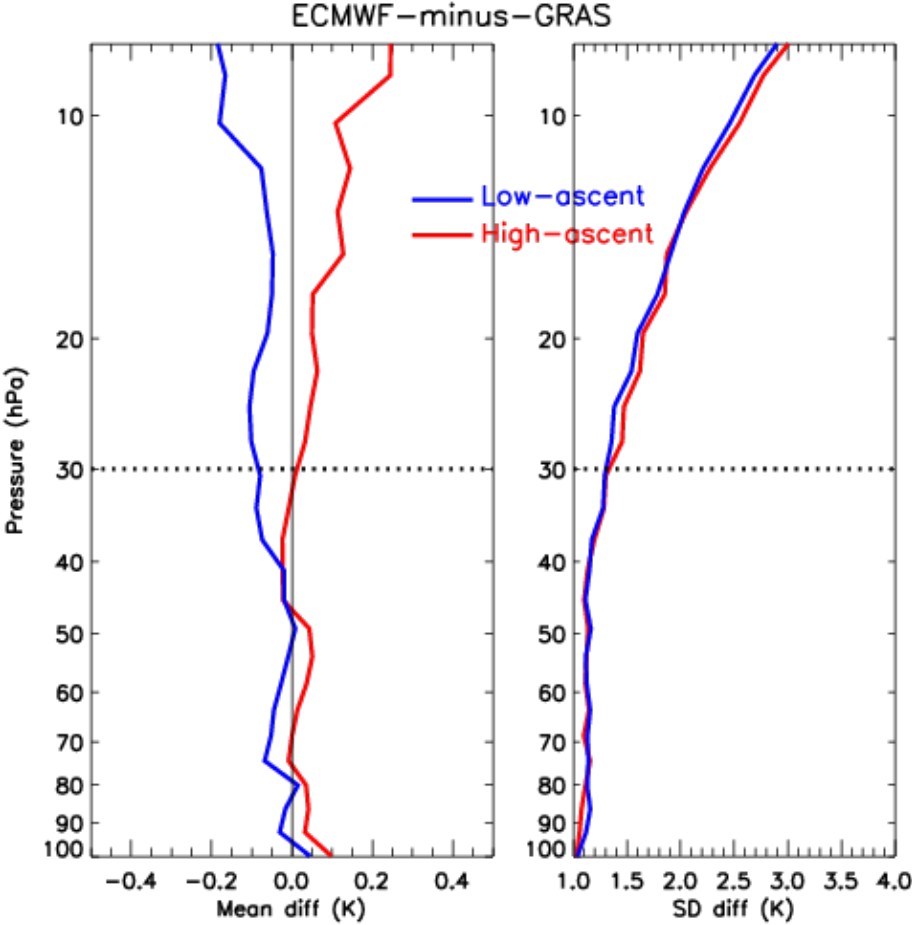

**Figure 18: As for Figure 16 but for difference of ECMWF-minus-GRAS Tdry. The number of collocated profiles for "High-ascent" and "Low-ascent" are ~1230 and ~1100, respectively.**



## 6 Summary and Conclusions

This paper discussed several scientific motivations for the high ascent attainment of upper-air measurements, up to 10–5 hPa (~32 km to ~37 km), rather than 30 hPa (~24.5 km) or lower altitudes, for balloon-borne radiosondes launched by GRUAN and other operational sites. The discussion included technical considerations for balloons and sensors (Section 2), and aspects from climate monitoring and process studies (Section 3), satellite validation including radiative transfer calculations (Section 4), and impacts on numerical weather forecasts (Section 5).

In general, larger-size balloons (e.g. 600–800 g for the case of a single radiosonde) reach higher burst heights, in addition to several simple measures including careful handling of the balloons when filling them with gas. Extremely cold temperatures found at e.g. nighttime tropical tropopause and polar-winter lower stratosphere, without solar heating, are often the cause of undesired early balloon burst. There are some proven techniques to overcome such situations, including the so-called kerosene treatment and storing balloons in a warm storage before launch.

There are various radiosonde sensors and special instruments that are flown together with a radiosonde, and some need improvements in measurement quality in the pressure range of 30–5 hPa. Modern GPS radiosondes can measure height and horizontal winds over the entire balloon coverage. Solar radiative heating in daytime soundings is the most important error source for the radiosonde temperature measurements, but recent developments in sensor characterization and data products, in particular by GRUAN, show that modern radiosondes have the potential to meet e.g. current WMO OSCAR requirements for uncertainties of atmospheric temperature measurements.

The climate and weather processes in the pressure (height) region 30 hPa to 5 hPa (~24.5 km to ~37 km) were reviewed in Section 3, by considering the role of radiosonde measurements there. Long-term cooling trends have been observed at least for the past 40 years in this region due primarily to the increase of greenhouse gases with modulations by evolving ozone changes. Radiosonde data products, being bias-adjusted using homogenization techniques, have been among the key data sets to quantify these trends, together with satellite data products from microwave-sounders and GNSS RO measurements. Radiosondes are also one of the very few sources of horizontal wind measurements in the stratosphere at the global scale and with a high vertical resolution. It is important to monitor the QBO in the tropics as it has global impacts through teleconnections. This can be done directly through high-altitude radiosondes, or indirectly through GNSS-RO and other observations. In the polar stratosphere, SSWs are a dramatic warming phenomenon in winter, which comes with sudden deceleration of the climatological westerly circulation of the polar vortex in association with planetary wave activity and has several remote effects both above and below the stratosphere, including surface weather and its predictability. Satellite temperature measurements together with thermal wind relationship characterize temperature and wind variations due to SSWs in data assimilation systems





reasonably well, but radiosonde data still provide important ground truth. It was also found that radiosonde soundings are still a key tool to investigate atmospheric gravity waves in the height region from ~25 km to ~40 km. Finally, some other processes requiring radiosonde measurements in the stratosphere were noted. Examples included the January 2022 eruption of Hunga Tonga-Hunga Ha'apai submarine volcano, where the Southern Hemisphere operational radiosonde network played a key role, providing unique information on the evolution and transport of the volcanic water vapour layers in the stratosphere during the first three months after the eruption.

Radiosonde measurements have always played an important role in CAL/VAL of various satellite measurements, ensuring the quality of satellite measurements, thereby enhancing the benefit of space programmes for Earth observations (Section 4). In Section 4.1, we provided cases where radiosonde profiles are directly fed into radiative transfer models to compare top-of-atmosphere simulated radiances with the corresponding satellite MW and IR observations, for quantification of the impact of radiosonde high-altitude attainment. The results for both MW and IR regions showed the importance of radiosonde high-altitude ceilings for validation of temperature sounding channels with peak sensitivities at these altitudes. GNSS RO measurements, either bending angles or refractivity data, have become increasingly important temperature information for both their accuracy and number of profiles per day since the 2000s. They are directly assimilated in recent operational analysis systems and reanalysis systems, but are also retrieved as temperature data by several institutes. The latter temperature retrieval products may differ in the stratosphere in particular above ~25 km due to differences in the processing algorithms (e.g. treatments of the ionospheric effects). In Section 4.2, we compared five GNSS RO temperature retrieval products with collocated RS92 GRUAN data products at Lindenberg, Germany and at Tateno, Japan, showing the value of reference-quality radiosonde temperature data above 30 km to validate GNSS RO temperature retrievals.

In Section 5, we evaluated the impacts of radiosonde measurements in the 30–5 hPa region on state of the art NWP systems. Section 5.1 explained the "anchor" atmospheric observations in the NWP systems, to correct for forecast model biases, which include radiosondes, GNSS RO, and some satellite radiance observations. In Section 5.2, we presented the results from radiosonde denial experiments using a Met Office operational global NWP system. In an experiment where radiosonde profiles truncated at 30 hPa, we found that a metric, the relative change in RMSE in the analysis against independent radiosondes at 10 hPa, shows a degradation for both temperature and wind (the impact is nearly twice as large for wind), depending on latitude bands, with the largest degradation observed in both cases in the winter hemisphere. For comparison, in an experiment where all GNSS RO data are removed, the same metric for temperature was of the same order with the largest degradation in the tropics, and that for wind was much smaller. We also found that the bias-correction coefficients have started drifting away from those in the control when radiosondes were not used to anchor the bias correction, indicating the role of high ascent radiosonde data in keeping the current quality of both analysis and forecasts through anchoring satellite measurements, although the current GNSS RO observations have a greater contribution. Finally, in Section 5.3, we tried to investigate the role of radiosonde data above the 10 hPa level (rather than the 30 hPa level in Section 5.2) in the NWP system by comparing the



NWP analysis fields from ECMWF associated with high-ceiling radiosondes with those associated with low-ceiling radiosondes. It was found that the temperature analysis associated with high-ceiling radiosondes tends to be warmer than the ones associated with low-ceiling radiosondes, their difference increasing (slightly) with height in the low-mid stratosphere, reaching 0.1–0.3 K at 10 hPa. Further study is needed to understand what is the impact of that difference on short-term weather forecasting and long-term climate change detection.

In conclusion, we showed that radiosonde measurements in the 30–5 hPa region are still very important even in the era of abundant satellite observations. Also, the future is uncertain in terms of both our observing system and our climate system under global warming. Our analysis strongly supports the contention that the extra costs and technical challenges involved in consistent attainment of high ascents are more than outweighed by the benefits for a broad variety of real time and delayed mode applications. Radiosonde stations across the broader observational network should pursue ceiling heights up to the 5 hPa level, especially sounding stations affiliated with dedicated networks such as GRUAN, GUAN, and GBON..

**Data availability**

Balloon burst point altitude data at various GRUAN sites are available upon request to the authors at GRUAN Lead Centre. Radiosonde data are available from https://www.gruan.org/data/data-products (last access: 24 May 2024) for the GRUAN data products (GDPs), https://cds.climate.copernicus.eu/cdsapp#!/dataset/insitu-observations-igra-baseline-network?tab=overview (last access: 31 May 2024) for RHARM and IGRA, and also from https://www.ncei.noaa.gov/products/weather-balloon/integrated-global-radiosonde-archive (last access: 31 May 2024) for IGRA. Also, BUFR radiosonde data as received at ECMWF are available from https://www.ncei.noaa.gov/data/ecmwf-global-upper-air-bufr/ (last access: 31 May 2024). Global atmospheric reanalysis data are available from https://cds.climate.copernicus.eu (last access: 24 May 2024) for ERA5 and https://doi.org/10.20783/DIAS.645 for JRA-3Q. Data from the radiative transfer calculations shown in Section 4.1 are available upon request to authors Domenico Cimini, Salvatore Larosa, and Lori A. Borg. GNSS RO temperature retrieval data are available from the URLs shown in Table 1. Data for Section 5.2 are available upon request to the authors at Met Office. Data for Sections 4 and 5.3 are available upon request to authors Bomin Sun and Anthony Reale.



**Author contributions**

PT and MF conceptualized the paper, and MF drafted Sections 1 and 6. CvR, MS, RD, MM, HV, RK, DE, KS, and MF drafted Section 2. FM, RJK, BD, and MF drafted Section 3. FC, AR, and MF drafted Section 4, DC, SL, and LB drafted Section 4.1, and N and MF drafted Section 4.2. FC, OL, BC, and CT drafted Sections 5.1 and 5.2, and BS and AR drafted Section 5.3. All the authors reviewed the draft and contributed to improvements of the manuscript.

**Competing interests**

At least one of the (co-)authors is a member of the editorial board of Atmospheric Measurement Techniques.
The authors have no other competing interests to declare.

**Acknowledgements**

MF was supported by JSPS KAKENHI grant nos. JP22H01303 and JP24K00700. DC and SL were supported by EUMETSAT study VICIRS (contract EUM/CO/22/4600002714/FDA). RJK was supported by NASA grant no. 80NSSC21K1968. BD was supported by NOAA-EPP grant no. NA22SEC4810015. N was supported by RIIM LPDP grant no. B-4038/III.4/FR.06/11/2023. PT was supported by Co-Centre award number 22/CC/11103. The Co-Centre award is managed by Science Foundation Ireland (SFI), Northern Ireland's Department of Agriculture, Environment and Rural Affairs (DAERA)
and UK Research and Innovation (UKRI), and supported via UK's International Science Partnerships Fund (ISPF), and the Irish Government's Shared Island initiative. HV's contribution is supported by the U.S. National Science Foundation (NSF) National Center for Atmospheric Research, which is a major facility sponsored by the NSF under Cooperative Agreement No. 1852977. We thank Junhong (June) Wang, Hannu Jauhiainen, Giovanni Martucci, Erik Andersson, Seiyoung Park, Adam Scaife, Roger Saunders, Warren Tennant, Neill Bowler, Chiaki Kobayashi, Shuhei Maeda, Chi O. Ao, Xavier Calbet, Axel
von Engeln, and John Eyre for valuable comments and suggestions on early versions of the manuscript. Use of the RHARM data as stated in the Copernicus license agreement is acknowledged.

**Financial support**

This work was supported by Co-Centre award no. 22/CC/11103 (Co-Centre award is managed by Science Foundation Ireland (SFI), Northern Ireland's Department of Agriculture, Environment and Rural Affairs (DAERA) and UK Research and



Innovation (UKRI), and supported via UK's International Science Partnerships Fund (ISPF), and the Irish Government's Shared Island initiative), JSPS KAKENHI grant nos. JP22H01303 and JP24K00700, EUMETSAT study VICIRS (contract EUM/CO/22/4600002714/FDA), NASA grant no. 80NSSC21K1968, NOAA-EPP grant no. NA22SEC4810015, RIIM LPDP grant no. B-4038/III.4/FR.06/11/2023.

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
