# Peer review of "Justification for high ascent attainment for balloon radiosonde soundings at GRUAN and other sites"

_EGUsphere, 2024_

## Author Comment (AC1)

Response to Anonymous Referee #1

Thank you very much for your review comments.

< General comments >

This article reviews and verifies the importance of radiosonde observation data in higher altitude beyond 10hPa with several demonstrations including numerical experiment. In general, it is well written and the article concept is clear. Numerical experiments described in Section 5.2 demonstrated well for the importance of higher altitude observation needs.

Thank you for your evaluation.

However, I wonder why this paper does not explicitly mention or provide any necessary (or desired) quantities for climate research. I mean there is no description about necessary accuracy, observation frequency, geographical distribution and so on. Of course, I can easily guess that it depends on the targets, so it might be difficult to identify or answer. In addition, actually, as to the accuracy, the authors referred the WMO's statement related to GBON (L. 97-109) and OSCAR requirements, and this article notes the recent radiosondes are satisfied with this. But, I wonder such accuracy should be considered in conjunction with temporal/spatial distribution in terms of climate research (or weather forecast).

Apologies for not mentioning about necessary accuracy, observation frequency, and geographical distribution for upper air observing networks in the submitted manuscript. We agree that including these points would make the paper more complete for such networks.

For example, Whiteman et al. (2011) made quantitative discussions on the relative importance of measurement accuracy and observation frequency in detecting trends for the case of upper tropospheric water vapor. Their methodology can be applied to other upper air variables in other regions of the atmosphere. Also, Weatherhead et al. (2017) made quantitative discussions on the geographical distributions (or more specifically the representativeness of a station and of a network of stations), using the GRUAN for illustration purposes, for both trend detection and climatology evaluation for the case of upper tropospheric temperature. Again, the methodology by Weatherhead et al. can be applied to other variables in other regions. Moreover, in the work by SY et al. (2021), temporal sampling effects on trend estimation have been quantified starting from real measurements and by

artificially reducing the size of the IGRA dataset, showing the impact at different pressure levels although without a reference to compare with.

The current paper focuses on the height attainment for radiosonde soundings, which were not fully discussed before in the peer-reviewed literature and which require renewed focus in an era of cost-savings when balloons may be the first line of budget cuts. Around the end of the Introduction section, we will add a new paragraph noting the above points so that interested researchers are informed of these published works and these broader considerations and can find out more if interested. We note that these topics are also under consideration by GRUAN as key scientific topics to address also to improve the requirements for GRUAN stations in the future and subsequent publications on these topics may be envisaged.

References:

SY, S., Madonna, F., Rosoldi, M., Tramutola, E., Gagliardi, S., Proto, M., Pappalardo, G.: Sensitivity of trends to estimation methods and quantification of subsampling effects in global radiosounding temperature and humidity time series, Int. J. Climatol., 41, 1– 23, https://doi.org/10.1002/joc.6827, 2020.

Weatherhead, E. C., Bodeker, G. E., Fassò, A., Chang, K.-L., Lazo, J. K., Clack, C. T. M., Hurst, D. F., Hassler, B., English, J. M., and Yorgun, S.: Spatial coverage of monitoring networks: A climate observing system simulation experiment, J. Applied Meteorol. Climatol., 56, 3211–3228, https://doi.org/10.1175/JAMC-D-17-0040.1, 2017.

Whiteman, D. N., Vermeesch, K. C., Oman, L. D., and Weatherhead, E. C.: The relative importance of random error and observation frequency in detecting trends in upper tropospheric water vapor, J. Geophys. Res., 116, D21118, https://doi.org/10.1029/2011JD016610, 2011.

In short, I'd like to know the goal that the authors aim by attaining data in higher-altitudes.

For example, as for the accuracy, the WMO's statement and/or OSCAR are the basis? The current GRUAN stations can provide enough number for the latter two (frequency, location)? Since it is expected that this article can act as a basis for future higher-altitude radiosonde soundings and this can solicit not only operational sites but also research sites to do such

observations , information or any suggestions are helpful for their justification.

The OSCAR requirements defined by WMO are a summarized set of reference values for the quality of measurements of atmospheric parameters, based on expert knowledge. They in particular cover the upper troposphere and lower stratosphere region and provide criteria for the Atmospheric Climate Forecasting and Monitoring application, i.e. the area on which GRUAN places emphasis. We therefore think that the WMO OSCAR requirements, in particular those for measurement uncertainties, are useful as orientation values for developing instruments and data products for upper air measurements, assuming that for the product delivered by a reference network, the most exacting (breakthrough) requirements set by OSCAR should be met. Regarding the GRUAN, we are still making significant efforts to expand the network, in particular in South America, Africa, and islands in the oceans, by recruiting new candidate sites (please see https://www.gruan.org/network/sites). Unfortunately, it has not yet reached the network of "35 to 40" stations that Seidel et al. (2009) proposed. Regarding the measurement frequency, we believe that the requirements detailed in the GRUAN Guide (WMO, 2013; see its Section 5.2. Levels of GRUAN operation), elaborated upon lessons learned by many scientists in different fields and part of the GRUAN community, as well as in line with the publications mentioned above, are still valid. However, as mentioned above, a separate study based on the existing GRUAN data holding is under consideration by the GRUAN community.

We would note that the basis behind the paper was more prosaic in nature than perhaps the reviewer has assumed. Site operators, both at GRUAN stations and at other stations, repeatedly requested justification as to why ascents should reach such exacting heights given the non-negligible costs involved in doing so. They and we were frustrated by the lack of a clear exposition of the value statement with the necessary rigorous basis. The paper deliberately therefore tries to investigate from several distinct angles the value proposition of high ascent attainment.

References:

Seidel, D. J., Berger, F. H., Diamond, H. J., Dykema, J., Goodrich, D., Immler, F., Murray, W., Peterson, T., Sisterson, D., Sommer, M., Thorne, P., Vömel, H., and Wang, J.: Reference Upper-Air Observations for Climate: Rationale, Progress, and Plans, Bull. Amer. Meteorol. Soc., 90, 361–369, https://doi.org/10.1175/2008BAMS2540.1, 2009.

WMO: The GCOS Reference Upper-Air Network (GRUAN) guide, Version 1.1.0.3, WIGOS Tech. Rep. 2013-03, GCOS-171, 116 pp., available at: https://www.gruan.org/documentation/gcos-wmo/gcos-171 (last access: 14 March 2025), 2023.

Below are specific comments, but I believe most of them are minor ones.

< Specific comments >

L. 253-273; In order for accurate description, it is better to mention that there are two types of wind measurements using GNSS/GPS; one measures the wind using the Doppler shift of the GPS frequency, and the others calculate it from difference of locations between the measured intervals.

Thank you for pointing this out. Instead of adding these details, we will simply write as, "to measure geometric height and horizontal winds."

L. 447-452; It might be useful to refer the recent hot topic "relationship between QBO and the Madden-Julian Oscillation," as it demonstrates stratospheric phenomenon (QBO) might control tropospheric phenomenon (MJO), which has great impact for the global weather forecast.

Thank you for this suggestion. We will add the following sentence, "In addition, the QBO may be modulating the Madden Julian Oscillation which is tropical tropospheric intra-seasonal variability (see e.g. Haynes et al., 2021, Section 3, for a review)."

Reference:

Haynes, P., Hitchcock, P., Hitchman, M., Yoden, S., Hendon, H., Kiladis, G., Kodera, K., and Simpson, I.: The Influence of the stratosphere on the tropical troposphere, J. Meteorol. Soc. Jpn., 99, 803–845, https://doi.org/10.2151/jmsj.2021-040, 2021.

L. 585; Please spell out "RO" as it appears first time here.

"radio occultation (RO)" first appears in the Introduction, in its second paragraph.

L. 634-635; After all, this accuracy is enough or not for any climate research?

The sentence at lines 634-635 indicates that CAL/VAL activities using NWP can detect biases that are larger than the overall uncertainty between NWP and GRUAN (~1–3 K range for humidity sounding channels and ~0.1–0.4K for temperature sounding channels). This is of the same order of magnitude as the user requirements for absolute calibration of current and future operational MW (e.g., 1.0–1.5 K for Metop-SG MWI and ICI channels) and IR sensor (e.g., 0.25 K for Metop-SG IASI-NG) sensors.

We will clarify the text by adding the following italic words to line 632:
the instrumental radiometric accuracy requirements of 0.25 K *in brightness temperature space*

L. 690; "These show ⋯ below 10hPa." should be inserted not here in this caption but in the text.

The sentence will be moved to the text.

Figs. 11 and 12; I wonder if only datasets, which contain higher (>30km) data, are used(*), mean and SD show similar tendency? Here, "similar" means for profiles above/below 30km. I thought It is helpful to know whether any effect from sampling exists or not. (*) ⋯ Namely, it means the calculation with the same number of pairs in the entire range (heights).

As shown in the following Figure R1, we see the same tendency if we only use the radiosondes reaching 35 km. We will add one sentence to the text, "Note that we see the same tendency for the Lindenberg case when we use radiosondes that reach 35 km only (not shown)."

[Figure]

Figure R1: As for Figure 11 (at Lindenberg) of the manuscript, but for the radiosondes reaching 35 km only. The number of collocation pairs for each data set is shown on the right.

L. 836-839; Isn't it possible to express "high-" in quantitatively?

Eyre (2016) and Francis et al. (2023) (both cited in the original manuscript) do not quantify high-quality or high-spatial or temporal coverage. For spatial and temporal coverage, they highlight the need for the data to be from a wide range of locations to capture diverse atmospheric conditions and reduce the risk of model bias contamination and stress the importance of continuity to track changes in atmospheric conditions and maintain the accuracy of bias corrections. We could consider, for example, that modern geostationary sounders providing sub-hourly observations every few kilometres can be considered as providing high-spatial (horizontal) and temporal coverage. As for the high-quality, here we are referring to observations with biases significantly smaller than model biases (such as radiosonde or RO measurements) such as they do not need to undergo a bias correction in the variational data assimilation systems.

L. 887-891; Could you explain a little bit more about "overall"? For example, if large RMSE is found in temperature for ECMWF (but others are small) and in wind for observations, they are evaluated on the same level?

The evaluation suite at the Met Office serves as a verification tool for assessing various aspects of scientific NWP data assimilation experiments. It calculates the change in root mean square forecast error (as a percentage) for key forecast variables at different atmospheric heights. These forecast errors are compared against independent references such as observations or ECMWF analysis. The mean change in forecast error averaged across all the variables such as $\Delta$RMSE = 100 × (RMSEControl – RMSETrial) / RMSEControl is what is referred to as overall change in RMSE in the text. As there is no weight given to the variables or forecast times, they all contribute equally to the overall score. Details of individual variables and forecast times can be visualized on 'scorecards' as shown below in Figure R2 and any discrepancies, say between verification against ECMWF and verification against observations, investigated further. For the experiments presented in the manuscript, good agreement was found between the verification against ECMWF and that against observations. Finally, if one or more variables stands out, detailed analysis can be conducted as presented in Figure 13 in the manuscript.

[Figure]

Figure R2: Change in root mean square forecast error for NS experiment verified using ECMWF analyses (left) and observations (rigth). Forecast degradations (relative to the control run) are denoted by downward triangles, whilst shading denotes statistical significance.

Figs. 15 and 17; Did the authors confirm any geographical bias for lower/higher soundings to avoid any possible their effect? (I mean, for example, Case A frequently does not appear in any latitudinal band?) Also, any bias for sampling time zone (00Z or 12Z) for the Cases A and B?

Differences which may be possibly present in geographic and time sampling between Case A and Case B have no impact on the statistics shown in Figures 16 and 18. This can be seen from the profiles of standard deviations for Case A and B in these figures, which are pretty much the same. Those standard deviations were computed by using the data from all locations and times. Thus, overall, the spatial and time sampling differences between Case A and B, if exist, have practically no impact on those statistics shown in Figures 16 and 18. We will add the following sentences in the revised text.
At the end of 4th paragraph of Section 5.3:
"It is noted that the profiles of standard deviation for Cases A and B in Figure 16 are quite similar, suggesting that spatial and temporal sampling differences between the two cases have practically no impact on the statistics shown in these figures."
At the end of 5th paragraph of Section 5.3:
"Also, the profiles of standard deviation for Cases A and B in Figure 18 are again quite similar."

L. 1120; (Middle) should be (Right).

Will be corrected.